

# Stand age and species composition effects on surface albedo in a mixedwood boreal forest

**Mohammad Abdul Halim**[1,2]**, Han Y. H. Chen**[3]**, and Sean C. Thomas**[1]

[1]Faculty of Forestry, University of Toronto, CE1 33 Willcocks Street, ON M5S 3B3, Canada
[2]Dept. of Forestry and Environmental Science, Shahjalal University of Science and Technology, Sylhet-3114, Bangladesh
[3]Natural Resources Management, Lakehead University, 955 Oliver Road, Thunder Bay, ON P7B 5E1, Canada

**Correspondence:** Mohammad Abdul Halim (abdul.halim@mail.utoronto.ca)

**Abstract.** TS1 Surface albedo is one of the most important processes governing climate forcing in the boreal forest and is directly affected by management activities such as harvesting and natural disturbances such as forest fires. Empirical data on the effects of these disturbances on boreal forest albedo are sparse. We conducted ground-based measurements of surface albedo from a series of instrument towers over 4 years in a replicated chronosequence of mixedwood boreal forest sites differing in stand age (to 19 years since disturbance) in both post-harvest and post-fire stands. We investigated the effects of stand age, canopy height, tree species composition, and ground vegetation cover on surface albedo through stand development. Our results indicate that winter and spring albedo values were 63 % and 24 % higher, respectively, in post-harvest stands than in post-fire stands. Summer and fall albedo values were similar between disturbance types, with summer albedo showing a transient peak at ∼ 10 years stand age. The proportion of deciduous broadleaf species showed a strong positive relationship with seasonal averages of albedo in both post-harvest and post-fire stands. Given that stand composition in mixedwood boreal forests generally shows a gradual replacement of deciduous trees by conifers, our results suggest that successional changes in species composition are likely a key driver of age-related patterns in albedo. Our findings also suggest the efficacy of increasing the proportion of deciduous broadleaf species as a silvicultural option for climate-friendly management of the boreal forest.

## 1 Introduction

Surface albedo, the fraction of incoming solar energy reflected from the surface in all directions, is one of the most important biophysical factors affecting both local and global climates. In boreal forest, the magnitude of albedo-related forcing on climate is even more important than in other ecosystems because of snow-related feedbacks, low sensible heat flux, and the relative stability of the atmospheric temperature profile due to weak latent-heat-driven convection (Bright et al., 2015b; Hansen et al., 2005). Even though albedo is increasingly used as an important state variable in climate models (Brown and Caldeira, 2017; Bala et al., 2007; Betts, 2000), forest-disturbance effects on net radiative forcing due to local albedo changes and related feedbacks with regional/ CE2 global mean surface temperature remain highly uncertain (Bright et al., 2015b; Lee et al., 2011). Harvest and fire suppression may differ substantially in their effects on albedo, but empirical data on albedo responses to disturbance type remain particularly sparse.

Following disturbance events, albedo of boreal forests is expected to change with stand age due to changing surface properties, and forest structure and composition. Age-related stand structural attributes (e.g., tree species composition, leaf area index (LAI), canopy height, and ground vegetation cover) can substantially influence surface albedo of a stand throughout the year. Studies have generally found higher albedos in young stands than in mature stands in the boreal forest (Bright et al., 2015b; Kuusinen et al., 2014; Amiro et al., 2006b), but the dynamic patterns with stand age remain unclear. The main study using ground-based measurements fit functions describing a linear decrease in sum-

mer albedo, and an exponential decrease in winder albedo, with stand age (Amiro et al., 2006b); however, in both cases the variability in young stands (< 25 years) was much greater than in older stands and poorly described by fitted mod-
els. Early in stand development, boreal mixedwood forests are commonly dominated by deciduous broadleaf species (Madoui et al., 2015; Brassard and Chen, 2010; Johnstone et al., 2010), which have higher leaf and canopy reflectance than conifers (Lukeš et al., 2013b; Linacre, 2003), contribut-
ing to high summer albedo in young stands (Lukeš et al., 2013a; Betts and Ball, 1997). These deciduous species shed leaves in the winter, which increases canopy openness (lowers LAI) and allows snow albedo to dominate, contributing to the high winter albedo in young stands. Available data sug-
gest that at this stage both LAI and ground vegetation cover usually increase with stand age, depending on site quality and silvicultural practices (Amiro et al., 2006b; Uotila and Kouki, 2005). Low LAI can increase canopy background reflectance both in snow-covered and snow-free conditions, and thus can
contribute to the high albedos in young stands (Amiro et al., 2006b). LAI effects on albedo in young stands may be highly modulated by ground vegetation cover in the summer but probably not much in the winter as ground vegetation is generally leafless or covered with snow (Kuusinen et al., 2015;
Lukeš et al., 2013a; Betts and Ball, 1997). In conjunction with other factors, surface albedo tends to decrease with increasing canopy height (Hovi et al., 2016; Linacre, 2003). In the later stages of stand development, albedo is expected to saturate non-linearly as conifers dominate the stand and
canopy cover and stand attributes change gradually, but data describing this pattern remain sparse (Amiro et al., 2006b).

Harvesting and fire are the major stand-replacing disturbances in the boreal forest (Brassard et al., 2008). These disturbances may differentially affect surface albedo of post-
disturbance stands in complex ways by altering ground surface spectral properties, species composition, and stand structure (Lukeš et al., 2013a; Liu et al., 2005), but field data directly addressing this issue are essentially limited to a single study in Europe (Kuusinen et al., 2016). Structure and
composition of post-fire stands are generally more heterogeneous than post-harvest stands; for example, post-fire stands are more likely to show a bimodal vertical structure and a mixture of conifer and hardwood species during early stand development stages (Brassard and Chen, 2010; Chen et al.,
2009). Charcoal residues may also strongly reduce albedo in snow-free conditions in the first years following fire disturbances (Amiro et al., 2006b). Both charcoal effects and stand heterogeneity might be expected to reduce surface albedo in post-fire stands relative to post-harvest stands. However,
the magnitude of this difference in surface albedo might be less than expected due to the presence of legacy charcoals from historical fires in post-harvest stands (Hart and Luckai, 2013). Immediately after harvesting, the albedo of a postharvest stand can also be reduced because of the presence of
coarse woody debris (CWD) and high soil moisture content (Linacre, 2003). In the years following a disturbance event, CWD might be expected to further reduce albedo by becoming darker in color due to decomposition processes (Brassard and Chen, 2008) and plant colonization (Kumar et al., 2018).

Despite the important roles of stand age, and stand structure and composition as determinants of boreal forest albedo, field measurements are scarce (Kuusinen et al., 2014) and particularly limited for early stand ages that show high variability in surface properties (Bright et al., 2013). This has contributed to poorly constrained estimates of the local albedo changes on net global radiative forcing (Bright et al., 2015b). Although some recent studies (e.g., Luyssaert et al., 2018; Naudts et al., 2016) have incorporated vegetation structure and composition in albedo estimation for land surface models, scarcity of field measurements is still a challenge for proper attribution of boreal forest albedo in climate models (Li et al., 2016; Thackeray et al., 2019). Thus, to estimate the net change in surface temperature as a function of albedo change from deforestation in boreal forests, a number of climate models (e.g., Bala et al., 2007; Betts, 2000) have used a "biome replacement" approach (replacing boreal forests with grassland or agricultural land cover types) and approximated boreal forests' albedo as a single value from mature stands (∼ 60 years old). Early stand dynamics are reported to determine which mechanism, albedo vs. carbon storage, dominates the net forcing for the boreal forest (Kirschbaum et al., 2011). Simplifications in climate models that do not explicitly consider stand age and successional effects on albedo will likely result in strongly biased estimations of boreal forests' albedo over the rotation (harvesting or fire) period (Bright et al., 2018).

Temporal dynamics of stand albedo following disturbance events have critical implications to interactions between climate forcing, forest management, and disturbance regimes. For example, if harvested stands converge in albedo with older stands within a few years (a small fraction of total rotation length), forest management is expected to have little impact on albedo at the landscape scale. Conversely, slow recovery in albedo or persistent effects of harvest compared to natural disturbance would indicate that the forest management regime fundamentally alters albedo-related climate feedbacks. Better understanding of post-disturbance patterns and of the mechanisms that account for variation in albedo will not only enhance global climate models (e.g., by improving the land-surface model: Bright et al., 2018), but also help to design climate-friendly silvicultural practices (Astrup et al., 2018; Matthies and Valsta, 2016; Bright et al., 2015b). In the present study, we set up micrometeorological towers with pyranometers in a replicated chronosequence of postharvest and post-fire sites to study stand age, disturbance type, and species composition effects on albedo in a mixedwood boreal forest in northwestern Ontario, Canada. We focused on the early stand development (0–19 years post harvest), where dynamics is expected be most rapid and where ground-based data from boreal forests is most sparse. We hy-

pothesized (1) that post-fire stands would show lower albedo values than post-harvest stands as a consequence of stand composition, legacy structures, and fire residues; (2) that all stands would approach albedo values similar to mature stands within 20–25 years, soon after crown closure; and (3) that stands with higher dominance of deciduous broadleaf species would show higher albedo than conifer-dominated stands, with this effect being most pronounced under snow-covered conditions.

## 2   Materials and methods

### 2.1   Study area

The study was conducted in the boreal forest of the Lake Nipigon region (49.55° N and 89.5° W), Ontario, Canada, approximately 200 km north of Thunder Bay. A series of circular (10 m radius) chronosequence plots were established in the post-harvest (full-tree harvest) and post-fire stands in the study area. Three plots were set up in each of three cutblocks (in separate stands) harvested in 1998, 2006, and 2013. Selected stands were at least 5 ha in size, and plots were established at least 100 m from any older or taller stand to avoid edge effects. Recent (2013) post-fire stands were not present, so we set up three plots only in post-fire stands dating from 1998 and 2006 fire events (Fig. 1). Replicate stands were spatially interspersed to the extent feasible. For each of the 15 plots, albedo and stand attributes (stand age, percentage of deciduous broadleaf species, canopy height, and percentage of ground vegetation cover) were measured from July 2013 to June 2017.

The mesic mixedwood study area is dominated by jack pine (*Pinus banksiana* Lamb.), black spruce (*Picea mariana* (Mill.) BSP), white spruce (*P. glauca* (Moench) Voss), trembling aspen (*Populus tremuloides* Michx.), eastern white cedar (*Thuja occidentalis* L.), balsam fir (*Abies balsamea* (L.) Mill.), and paper birch (*Betula papyrifera* Marsh.) (Chen and Popadiouk, 2002). The management regime in the region is based on clearcut silviculture modified to include live tree retention in harvested stands (OMNRF, 2015); typical rotation lengths are 80 years (Colombo et al., 2005). In study plots over the study period canopy height ranged from 0–7.7 m, ground vegetation cover ranged from 1.8 %–96.7 %, LAI from 0–2.1, and the proportion of deciduous broadleaf basal area from 10.6 %–100 %, and stand density from 0–11556 stem ha$^{-1}$ (Table 1). The study area has an average elevation of 416 m a.s.l. The soil is a moderately deep Brunisol (coarse-loamy texture) with 1–15 cm thick organic layer (i.e., the total litter, fermented, and humic (LFH) layers). The area remains snow covered for 5–6 months with an average snow depth of $\sim$ 10 cm (Environment Canada, 2018; Sims et al., 1997), and the mean annual air temperature of the study plots was $-1.1$ °C (Halim and Thomas, 2018).

### 2.2   Experimental setup

In the center of each circular plot a pair of upward- and downward-facing pyranometers (silicon (Si) pyranometer; Onset, Massachusetts, USA; measurement range 0–1280 W m$^{-2}$ over a spectral range of 300–1100 nm, accuracy $\pm 5$ %, resolution 1.25 W m$^{-2}$) were set up on a mast 3.5 m above the canopy (above the ground for 2013 post-harvest stands) to measure incident and reflected solar radiation every 10 min. The plot and tower locations were selected to avoid trees from surrounding stands falling within the footprints of the pyranometers or blocking incoming solar radiation. Instrument masts consisted of extendable galvanized steel poles and were set in concrete bases and guyed to mitigate instrument sway. At least once a year pyranometer heads were cleaned and realigned to make sure they were normal to the ground. Average daily albedo was calculated as the ratio of daily total incident and reflected radiation for each plot. The average daily albedo was used to calculate average monthly albedo, which was finally used to calculate mean seasonal albedo for each year in each plot.

Quality control for the irradiance and reflected solar radiation measurements was conducted following guidelines of the World Meteorological Organization (WMO). Any unusually high or low values were replaced by interpolated values by taking the average of preceding and subsequent measurements. Daily total irradiance data were compared against the WMO-provided maximum possible daily sums of clear-sky irradiance for 50° N latitudes (WMO, 1987, p. 26). If the measured daily total irradiance was higher than the maximum possible value, we excluded the measurements for that day. For reflected solar radiation, if the daily total of reflected solar radiation was higher than the daily total irradiance, we also excluded the measurements for that day. In addition, we excluded measurements for any snowy day; snowfall was detected using data from the closest available weather station (Environment Canada, 2018).

In addition to albedo, winter (December–February)/spring (March–May) and summer (June–August)/fall (September–November) proportions of deciduous-broadleaf basal area (%), canopy height (m), and ground vegetation cover (%) were measured every year in late October and early July, respectively, in each plot. The proportion of deciduous broadleaf species (%) were determined for trees with diameter at breast height $\geq 5$ cm and height $> 1$ m. Canopy height was determined as the mean height of all trees sampled; the young stands sampled were at stages of development prior to and just after canopy closure, so essentially all trees were "canopy dominants". The proportion of deciduous broadleaf species of a plot was calculated as the ratio of basal area of the deciduous species to the total basal of area of the plot. In each plot, four 1 m$^2$ subplots were set up, and percent ground vegetation cover was determined visually (Kumar et al., 2018). Stand age was determined as the time (year) since the last disturbance

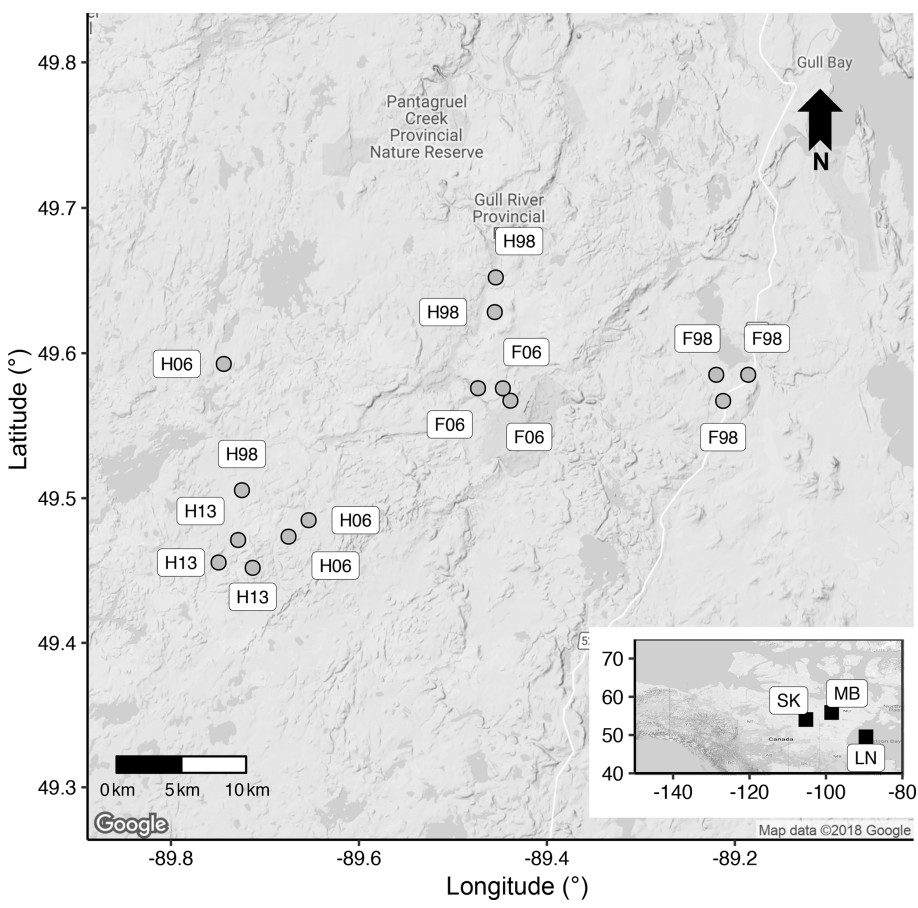

**Figure 1.** Map of the study area. Labels in the rectangular boxes indicate disturbance types (H is harvest and F is fire) and years (98: 1998; 06: 2006; and 13: 2013) for each plot (gray circles). Inset: black squares indicate locations of all data sources including the current study area (LN: Lake Nipigon area, Ontario, Canada; SK: Saskatchewan, Canada; MB: Manitoba, Canada).

**Table 1.** Structural characteristics of post-harvest and post-fire stands sampled. Mean values ($\pm$ SE) are reported across all sites of a given disturbance type.

| Stand type | Stand age (years) | DBS (%) | LAI | Stem density (stem ha$^{-1}$ $\geq 5$ cm DBH) | Height (m) | GCV (%) |
|---|---|---|---|---|---|---|
| Post-harvest | 0–19 | $55.4 \pm 11.2$ | $0.4 \pm 0.3$ | $6472 \pm 3060$ | $1.7 \pm 1.3$ | $51.8 \pm 20.1$ |
| Post-fire | 7–19 | $37.8 \pm 9.1$ | $0.7 \pm 0.4$ | $8400 \pm 1902$ | $2.9 \pm 1.5$ | $62.5 \pm 14.1$ |

Notes: DBS, LAI, and GCV indicate deciduous broadleaf species (% by basal area), leaf area index, and ground cover vegetation.

(fire or harvesting) for each plot. Fire maps (from the Ontario Ministry of Natural Resources, Canada) and forest management plans (from Resolute Forest Products, Canada) were used to verify type and year of disturbances.

**2.3 Sources of secondary data**

Since we did not have recent post-fire stands (0–6 years old) in the study area, we used secondary albedo data from studies in post-fire boreal forests with similar stand characteristics in Saskatchewan and Manitoba (Canada) (Fig. 1). We also used secondary albedo data for old stands ($> 70$ years)

from these sites along with primary data to develop regression models for both post-fire and post-harvest stands. Here we assumed that at this late stage of stand development, there is negligible difference in stand attributes (e.g., species composition, height, LAI) between post-fire and post-harvest stands (Moussaoui et al., 2016). We did not use satellite-based albedo data as secondary sources as they tend to diverge from field measurements depending on a number of factors including stand age, latitude, and cloud-cover effects (Halim and Thomas, 2017; Bright et al., 2015a).

Data for Saskatchewan sites were retrieved from Amiro et al. (2006a), and for Manitoba sites data were obtained CE4 from Amiro et al. (2006b) by digitizing data points from relevant figures using the WebPlotDigitizer software (Rohatgi, 2018). These stands were dominated by jack pine and black spruce with some intermixing of trembling aspen. All post-fire sites (including this study) had severe fires that completely killed previous vegetation. There were a few burned snags in the Saskatchewan and Manitoba sites and none in the present study sites. These areas remain snow covered for ∼ 6 months with average snow depths of 10–15 cm (Environment Canada, 2018). Pyranometers were located in Saskatchewan sites at 18–20 m and in Manitoba sites at 6 m heights. There was no detailed information on how proportions of broadleaf deciduous species were calculated for these sites; however, we assumed they were basal-area based. A detailed description of the study areas and methods can be found in the respective articles.

## 2.4 Accuracy assessment of albedo measurements

To test the relative accuracy of albedo measurements from Si-based pyranometers (Hobo, Onset Computers; used in this study) in comparison to thermopile pyranometers (CNR1, Kipp & Zonen; used in the studies providing secondary data), we conducted a field calibration study over 9 d under variable sky and ground conditions (see Supplement). Results from this study showed a very close agreement between the measurements of Hobo and CNR1 pyranometers (Fig. S1 in the Supplement). The difference (CNR1 – Hobo) in daily albedo over the study period ranged from $-0.0601$ to $+0.064$, and the mean difference in daily albedo was $0.0028$ ($\pm 0.031$). The mean difference was negligible and the range in differences was well within the previously reported error ranges (∼ 5 %–7 %) for similar pyranometers (Myers, 2010; Stroeve et al., 2005). We also did not observe any detectable pattern in deviations between the pyranometers under different sky and ground conditions. We therefore concluded that albedo measurements from Si-based Hobo and thermopile-based CNR1 pyranometers are closely comparable, and corrections to the Si-based albedo estimates presented are not warranted.

## 2.5 Measurements of ground surface reflectance

To examine effects of disturbance type on ground surface reflectance, three soil samples (top 10 cm including LFH layer, surface area 78.5 cm$^2$) from each plot were collected in fall 2017 to measure the ground surface reflectance. Samples were all collected within a 2 d precipitation-free period and were brought to the lab in airtight packaging without disturbing the top surface. Surfaces of these samples were visually assessed for presence of visible charcoal fragments. A spectrometer (SD 2000; Ocean Optics, Florida, USA; measurement spectral range 338.7–1001.8 nm) equipped with an integrating sphere was used to measure the directional hemispherical reflectance factor of the top surface of the soil samples. As there were no recent post-fire stands in the study area, we collected charcoal samples (of twigs, branches, barks, and stems) from the forest floor of a jack pine dominated post-fire (fire occurred in 2011) stand in summer 2015 from near the Musselwhite mine (52.61° N and 90.37° W), Ontario, Canada. Every sample was measured 10 times in 10 different locations (each 0.84 cm$^2$ in area), and each measurement was performed by scanning 10 times (with Boxcar width 5 (spatial averaging of 5 pixels) and 100 ms integration time) to get an average reflectance for each location of a sample. Details of the spectrometer and integrating sphere used can be found in the Materials and Methods section of Baltzer and Thomas (2005). Forest floor reflectance values from the Musselwhite stand (4 years old) were compared to soil sample reflectance values from recent (2013) post-harvest stands (4 years old). For older stands (1998 and 2006 post-harvest and post-fire stands), soil sample reflectance data were compared using samples from the main study plots.

## 2.6 Data analysis

Robust $t$ tests (Wilcox, 2016) were used to compare mean differences in ground surface reflectance (in visible (400–700 nm) and near-infrared (> 700–1000 nm) spectral bands) and seasonally averaged albedo between post-harvest and post-fire stands. Mean seasonal albedo values of post-harvest and post-fire stands were also compared using analysis of covariance (ANCOVA) controlling for the effects of stand age as a covariate. Secondary albedo data for 0–6-year-old post-fire stands were only available for winter and summer seasons. Therefore, in the $t$ tests (and in ANCOVA) for winter and summer albedo, data from 0–19-year-old post-harvest and post-fire stands were used. For spring and fall, albedo data from recent (0–6 years old) post-harvest stands were omitted (since there were no data from post-fire stands for these seasons), and data from 7–19-year-old stands were used to make the comparisons unbiased. Secondary data from old stands (> 70 years) were not used in the $t$ tests or ANCOVA. These analyses treat seasonally averaged albedo values from the same stands as independent. We also conducted parallel analyses using linear mixed models that included plot as a random variable; in all cases, the random effect was not significant, and thus only the simpler linear model results are presented.

Generalized linear models (GLMs) with the log-linked Gaussian family (additive-observation-error model with constant variance) were found to be the best fitted to model seasonal albedo as a function of stand attributes (stand age, proportion of deciduous broadleaf species, canopy height, ground vegetation cover, and their interactions) for both post-harvest and post-fire stands. Best models were chosen using an AIC CE5 based stepwise algorithm. Asymptotic chi-square statistics based on deviance were calculated for each best-fit

model to test if the model was significantly better than its counterpart null model. We could not use a GLM to predict fall albedo because some stand attributes were only nonlinearly (double exponentially) related to albedo. If we included these nonlinearly related stand attributes with other attributes in a GLM, the model structure became very complex (a mixture of nonlinear and linear families), and defining an appropriate GLM family became a statistical challenge. To avoid modeling complexity, for each of these nonlinearly related fall attributes a separate nonlinear model was fitted, and for other attributes GLMs with identity-linked Gaussian family were found be the most suitable. The $\Delta$AIC for each best-fit model is calculated as its AIC difference with the corresponding null model (AIC of the best-fit model − AIC of the corresponding null model). Sample-size-corrected AIC values were used in all cases. Using the identical model selection approach, we conducted a similar analysis of the dataset after excluding measurements from secondary sources; since the model outputs were similar, we present this analysis as a supplementary table (Table S1 in the Supplement).

Data were analyzed using the R platform (R Core Team, 2018) and graphs were prepared using the ggplot2 package (Wickham, 2016). Robust $t$ tests were done by 10 000 bootstrapped samples considering mean as an estimator for group comparison, and implemented by the *pb2gen* function of the WRS2 R package (Mair and Wilcox, 2018). Adjusted $R^2$ values for GLMs were calculated using the *rsq* function of the R package rsq (Zhang, 2018).

## 3 Results

### 3.1 Seasonal albedo in post-harvest and post-fire stands

Albedo differences between post-harvest and post-fire stands varied among seasons. Albedo values in periods of the year with appreciable snow cover were significantly higher in post-harvest stands than in post-fire stands (for winter: 0.56 vs. 0.34, $p < 0.01$; for spring: 0.32 vs. 0.24, $p = 0.11$). Summer albedo values were also marginally higher in post-harvest stands ($p = 0.24$), and fall albedos were similar between disturbance types ($p = 0.73$) (Fig. 2). Considering stand age as a covariate, ANCOVA results also indicate higher albedo of post-harvest stands in winter ($p = 0.02$), spring ($p = 0.15$), summer ($p = 0.04$), and similar in fall ($p = 0.77$) compared to post-fire stands. Data also suggest higher variability in albedo in post-harvest stands than in post-fire stands (Fig. 2).

### 3.2 Ground surface reflectance in post-harvest and post-fire stands

Surface charcoal fragments were visually observed in all post-fire soil core samples and in 70 % of post-harvest samples. Specular-included reflectance measurements of ground surface samples suggest that differences in ground surface

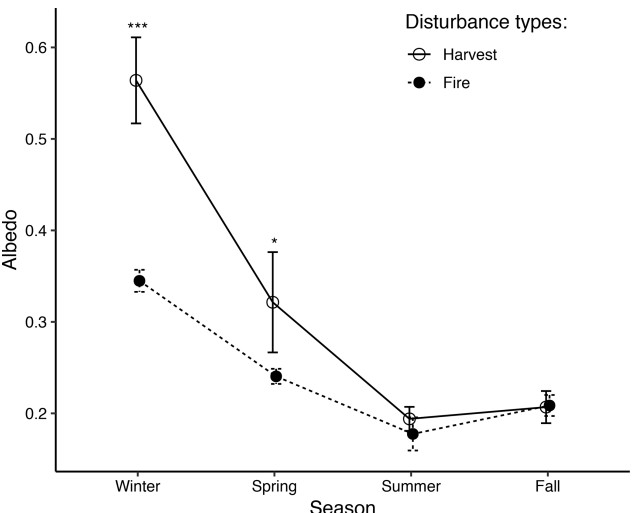

**Figure 2.** Comparison of seasonal albedo (mean $\pm$ SE) in post-harvest and post-fire stands. Winter (no. of observations for post-fire stands, $n_F = 35$; no. of observations for post-harvest stands, $n_H = 48$) and summer ($n_F = 44$, $n_H = 41$) albedo data were from 0–19-year-old stands, and spring ($n_F = 30$, $n_H = 30$) and fall ($n_F = 30$, $n_H = 30$) albedo data were from 7–19-year-old stands. Albedo of 0–6-year-old post-fire stands were from secondary sources. Symbols * and *** indicate significant mean albedo differences between post-harvest and post-fire stands with $p = 0.11$ and $p < 0.01$, respectively.

characteristics contribute to overall surface albedo in the study sites. Summer ground surface reflectance was generally higher in old stands (Fig. 3b) than in young stands (Fig. 3a) particularly in the 600–1000 nm range. Young (4 years old) post-harvest stands showed significantly lower mean ground reflectance values (74.3 %, $p < 0.01$) in the visual spectrum (400–700 nm) and higher (32.3 %, $p < 0.01$) in the near-infrared spectrum ($> 700$–1000 nm) than those of young post-fire stands (Fig. 3a). Older (11 and 19 years old) post-harvest stands however showed higher mean ground reflectance in both visible (31.7 %, $p < 0.01$) and near-infrared (4.6 %, $p < 0.01$) spectra compared to post-fire stands (Fig. 3b).

### 3.3 Seasonal albedo in relation to stand attributes in post-harvest and post-fire stands

### 3.3.1 Winter albedo in post-harvest and post-fire stands

Results from the best-fit GLM ($p < 0.01$, adj. $R^2 = 0.97$) for post-harvest stands indicated that stand age, proportion of deciduous broadleaf species, canopy height, and interactions among these variables were significant predictors of winter albedo (Table 2). Stand age was related to winter albedo via an exponential decay model with a horizontal asymptote ($\Delta$AIC $= -25.5$), and all estimated model parameters

**Biogeosciences, 16, 1–18, 2019** www.biogeosciences.net/16/1/2019/

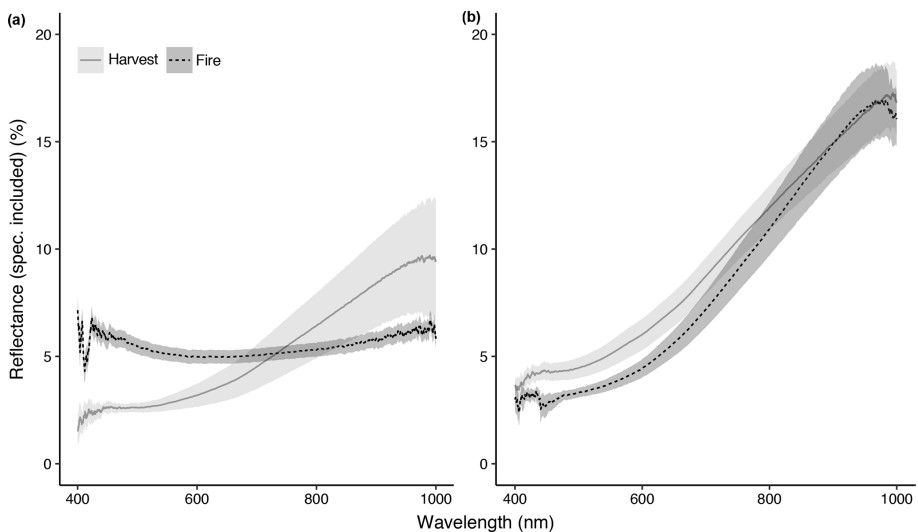

**Figure 3.** Specular-included ground surface reflectance (400–1000 nm) of post-harvest and post-fire stands. Lines indicate mean reflectance (number of sample ($n$) ×10 replicated measurements/CE6 sample) in the corresponding wavelengths, and shades indicate SE. **(a)** ground surface reflectance of young (4 years old) post-harvest stands ($n = 9$) and a post-fire stand ($n = 12$). **(b)** ground surface reflectance of old (11 and 19 years old) post-harvest ($n = 18$) and post-fire ($n = 18$) stands.

were significant (for 0.19 and 0.55: $p < 0.01$; for $-0.06$: $p < 0.05$) (Fig. 4a). The proportion of deciduous broadleaf species (Fig. 6a) and canopy height (Fig. 7a) were also related to winter albedo via negative exponential models with horizontal asymptotes ($\Delta$AIC $= -6.7$ and $-100.4$, respectively), and all estimated parameters for both models were significant ($p < 0.01$).

For post-fire stands the best-fit GLM ($p < 0.01$, adj. $R^2 = 0.75$) indicated that stand age and proportion of deciduous broadleaf species were significant predictors of winter albedo (Table 2). Stand age was related to winter albedo via an exponential decay model with a horizontal asymptote ($\Delta$AIC $= -38.5$), and all estimated model parameters were significant ($p < 0.01$) (Fig. 4b). The proportion of deciduous broadleaf species was related to winter albedo via a negative exponential model with horizontal asymptote ($\Delta$AIC $= -16.3$), and all estimated model parameters were significant (for $-0.27$: $p < 0.06$; for 1.02 and 0.45: $p < 0.01$) (Fig. 6b).

### 3.3.2 Spring albedo in post-harvest and post-fire stands

For post-harvest stands the best-fit GLM ($p < 0.01$, adj. $R^2 = 0.99$) indicated that stand age, proportion of deciduous broadleaf species, height, the interaction of stand age, CE7 and proportion of deciduous broadleaf species were significant predictors of spring albedo (Table 2). Stand age (Fig. 5a) and canopy height (Fig. 7c) were related to spring albedo via exponential decay models with horizontal asymptotes ($\Delta$AIC $= -15.1$ and $-31.2$, respectively). Estimated parameters of stand age-albedo (for 0.26: $p < 0.01$; for 0.72 and $-0.72$: $p < 0.05$) and canopy height-albedo (for 0.16 and 0.33: $p < 0.01$; for $-1.84$: $p = 0.07$) models were

likewise significant. The proportion of deciduous broadleaf species was related to spring albedo via a negative exponential model ($\Delta$AIC $= -6.72$), and all estimated parameters were significant ($p < 0.01$) (Fig. 6c).

The best-fit GLM ($p < 0.01$, adj. $R^2 = 0.99$) for post-fire stands indicated that stand age, and proportion of deciduous broadleaf species were the only significant predictors of spring albedo (Table 2). Stand age (Fig. 5b) and proportion of deciduous broadleaf species (Fig. 6d) were related to spring post-fire stand albedo via exponential negative growth models ($\Delta$AIC $= -7.0$ and $-7.5$, respectively), and all estimated parameters for both models were significant ($p < 0.01$).

### 3.3.3 Summer albedo in post-harvest and post-fire stands

The best-fit GLM ($p < 0.01$, adj. $R^2 = 0.97$) for post-harvest stands indicated that stand age, proportion of deciduous broadleaf species, ground vegetation cover and its interaction with stand age and proportion of deciduous broadleaf species were significant predictors of summer albedo (Table 2). Stand age alone (not with other stand attributes as in the GLM) was related to summer albedo via a double exponential model ($\Delta$AIC $= -73.1$), and all estimated model parameters were significant ($p < 0.01$) (Fig. 4c). The pattern described by this function indicates a sharp peak in albedo with a maximum at 10–15 years of stand age. The proportion of deciduous broadleaf species is related to summer albedo via a three-parameter sigmoid model ($\Delta$AIC $= -48.6$), and all the estimated parameters were significant ($p < 0.01$) (Fig. 6e). Ground vegetation cover was related to summer albedo via an exponential model with a Gumbel distribution without a

**Table 2.** Regression model coefficients and fit statistics for albedo as a function of stand attributes in different seasons in the boreal forest.

| Season | Post-harvest stands | | | | Post-fire stands | | | |
|---|---|---|---|---|---|---|---|---|
| | Parameter estimates | | Model fit | | Parameter estimates | | Model fit | |
| | Coefficient | Estimate | ΔAIC | Adj. $R^2$ | Coefficient | Estimate | ΔAIC | Adj. $R^2$ |
| Winter | Intercept | **1.772** | | | Intercept | **−1.25** | | |
| | SA | **−0.031** | | | SA | −0.004 | | |
| | PDBS | **−0.021** | −69.2 | 0.97 | PDBS | **0.005** | −5.3 | 0.75 |
| | CH | **−0.079** | | | | | | |
| | SA : CH | **0.002** | | | | | | |
| | PDBS : CH | **−0.007** | | | | | | |
| Spring | Intercept | **−7.195** | | | Intercept | **−1.747** | | |
| | SA | **1.298** | | | SA | **0.016** | | |
| | PDBS | **0.116** | −495.4 | 0.99 | PDBS | 0.002 | −18.8 | 0.92 |
| | CH | **−1.264** | | | | | | |
| | SA : PDBS | **−0.024** | | | | | | |
| Summer | Intercept | **−1.377** | | | Intercept | **−2.996** | | |
| | SA | **0.032** | | | SA | **−0.012** | | |
| | PDBS | **−0.003** | | | PDBS | −0.004 | | |
| | GVC | **−0.01** | −24.9 | 0.97 | CH | **0.788** | −48.3 | 0.95 |
| | SA : GVC | **−0.0004** | | | SA : PDBS | **0.003** | | |
| | PDBS : GVC | **0.0001** | | | SA : CH | **−0.004** | | |
| | | | | | SA : CH : PDBS | **−0.001** | | |
| Fall | Intercept | **0.398** | | | $\frac{4.5}{6.87}e^{\left(-\frac{SA-13.2}{6.87}-e^{-\frac{SA-13.2}{6.87}}\right)}$ | | −3.1 | 0.045[a] |
| | SA | 0.013 | | | $0.099e^{0.013\,\mathrm{PDBS}}$ | | −25.4 | 0.008[a] |
| | CH | **−0.182** | | | | | | |
| | GVC | **−0.007** | −6.1 | 0.94 | | | | |
| | SA : CH | **0.007** | | | | | | |
| | CH : GVC | **0.005** | | | | | | |
| | SA : CH : GVC | **−0.0002** | | | | | | |
| | $\frac{28.86}{45.39}e^{-\frac{PDBS-67.62}{45.39}-e^{-\frac{PDBS-67.62}{45.39}}}$ | | −0.9 | 0.049[a] | | | | |

Notes: SA, PDBS, CH, and GVC indicate stand age (year), proportion of deciduous broadleaf species (%), canopy height (m), and ground vegetation cover (%), respectively. Parameter estimates for GLMs in bold and regular fonts indicate statistical significance at the 1 % and 5 % levels, respectively. For fall nonlinear regression models, 28.86 and 45.39 coefficients of post-harvest stands are significant at the 5 % level, and the rest is significant at the 1 % level. [a] Indicates residual standard error of the nonlinear regression model. ΔAIC = AIC of the best-fit model − AIC of the corresponding null model.

horizontal asymptote (ΔAIC = −25.8), and all estimated parameters were significant ($p < 0.01$) (Fig. 8e).

For post-fire stands the best-fit GLM ($p < 0.01$, adj. $R^2$ = 0.95) indicated that stand age, proportion of deciduous broadleaf species, canopy height, and their interactions with stand age were significant predictors of summer albedo (Table 2). Stand age (Fig. 4d) and canopy height (Fig. 7f) were related to summer post-fire stand albedo via exponential models with Gumbel distributions with horizontal asymptotes (ΔAIC = −49.3 and −5.3, respectively). As in the case of post-harvest stands, peak albedo was found at ∼ 10–15 years of stand age. All estimated parameters of stand age-albedo and canopy height-albedo models were significant ($p < 0.01$). The proportion of deciduous broadleaf species was related to summer albedo via a negative exponential

growth model (ΔAIC = −6.8), and all estimated model parameters were significant ($p < 0.01$) (Fig. 6f). Two instances of particularly high summer albedo measurements ($> 0.2$) were found in aspen-dominated stands affected by damage from aspen serpentine leaf miner (*Phyllocnistis populiella*).

### 3.3.4 Fall albedo in post-harvest and post-fire stands

The best-fit GLM ($p < 0.01$, adj. $R^2$ = 0.94) for post-harvest stands indicated that stand age, canopy height, ground vegetation cover, and their interactions were significant predictors of fall albedo (Table 2). Proportion of deciduous broadleaf species was also an important predictor that was modeled separately via a double exponential model (and was not added to the GLM to avoid modeling complexities) (ΔAIC =

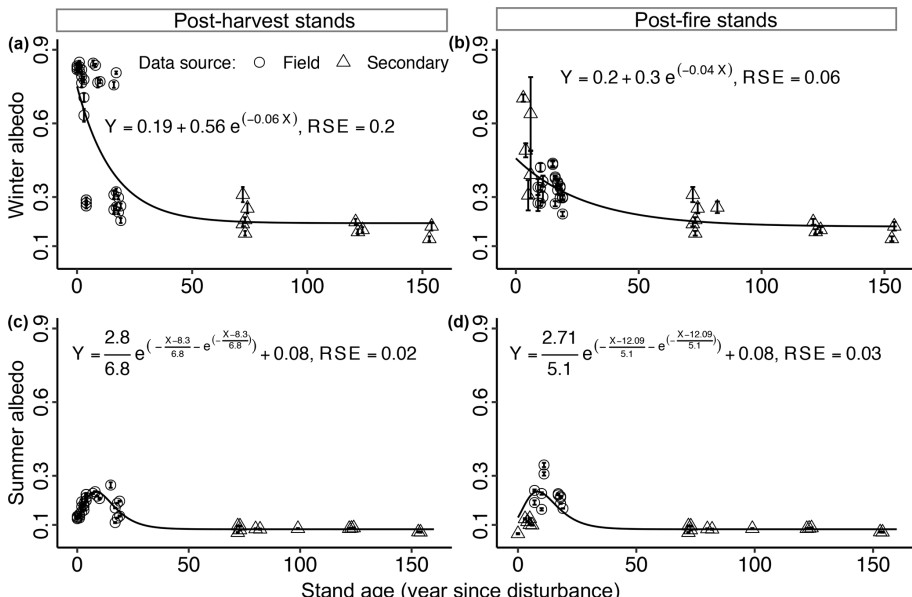

**Figure 4.** Stand age affecting mean seasonal albedo ($\pm$ SE) in boreal forest over 0–150 years of stand development. Mean winter albedo as a function of stand age in **(a)** post-harvest stands ($n = 42$) and **(b)** post-fire stands ($n = 36$). Mean summer albedo as a function of stand age in **(c)** post-harvest stands ($n = 41$) and **(d)** post-fire stands ($n = 30$). Each field-data point is the average seasonal albedo (error bars indicate standard errors) of three plots from each stand-age category over the study period.

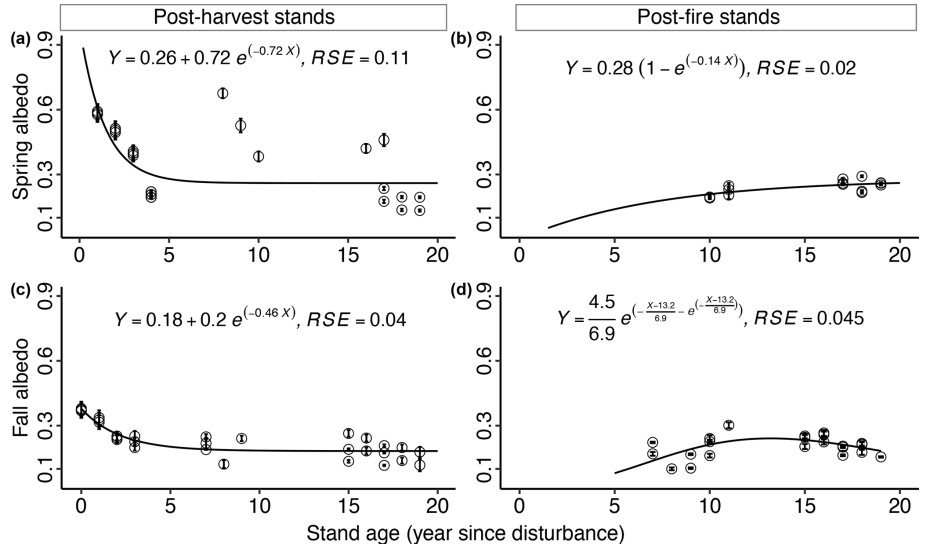

**Figure 5.** Stand age affecting mean seasonal albedo ($\pm$ SE) in boreal forest in the early seral stage. Mean spring albedo as a function of stand age in **(a)** post-harvest stands ($n = 26$) and **(b)** post-fire stands ($n = 14$). Mean fall albedo as a function of stand age in **(c)** post-harvest stands ($n = 29$) and **(d)** post-fire stands ($n = 22$). Each field-data point is the average seasonal albedo (error bars indicate standard errors) of three plots from each stand-age category over the study period.

$-0.9$); all estimated model parameters were significant (for 28.9 and 45.4: $p < 0.05$; for 67.6: $p < 0.01$) (Fig. 6g). Stand age (Fig. 5c) and ground vegetation cover (Fig. 8g) were related to albedo via exponential decay models with horizontal asymptotes ($\Delta$AIC $= -36.8$ and $-28.38$, respectively), and all estimated parameters for both models were significant ($p < 0.01$). Canopy height was also related to albedo

via a negative exponential model ($\Delta$AIC $= -11.2$), and all estimated parameters were significant ($p < 0.01$) (Fig. 7g).

To avoid modeling complexities, stand age and proportion of deciduous broadleaf species were fitted individually with fall albedo of post-fire stands (Table 2). Stand age was related to albedo via a double exponential model ($\Delta$AIC $= -3.1$), and all estimated model parameters were significant ($p <$

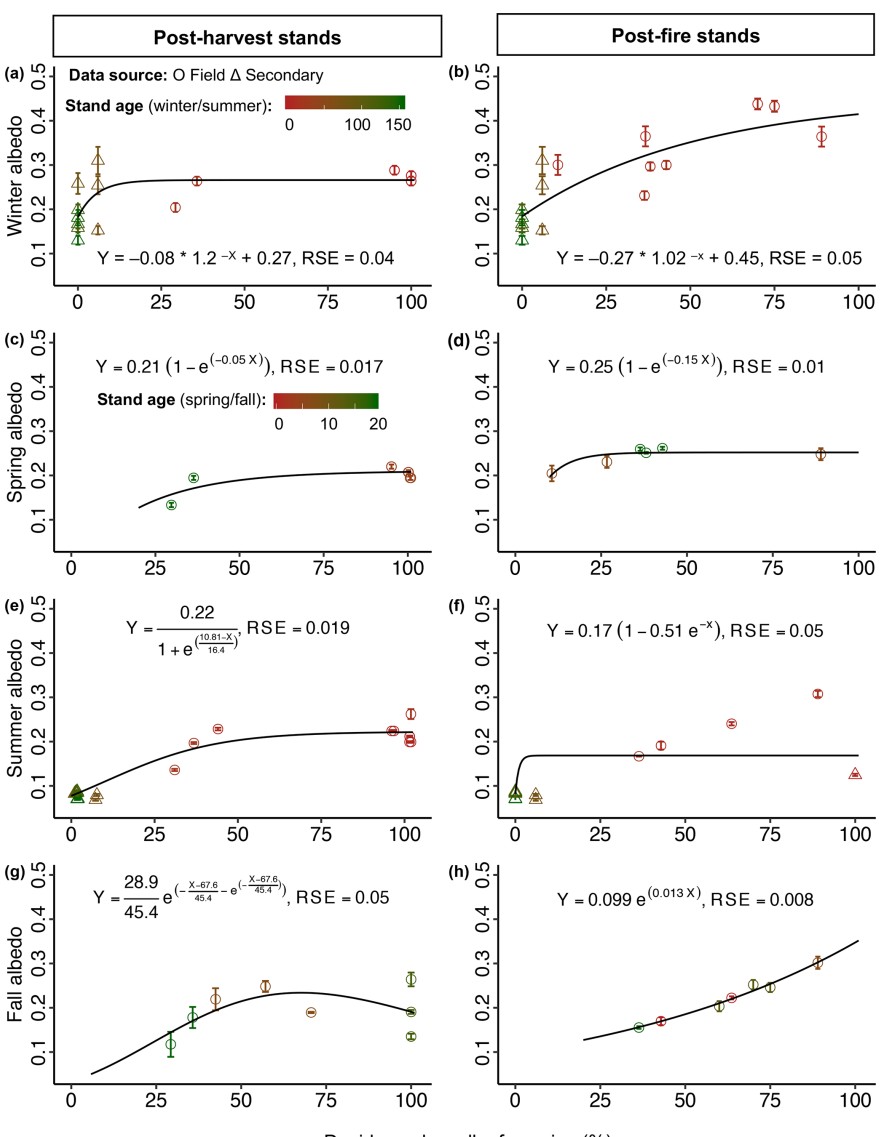

**Figure 6.** Mean seasonal albedo (±SE) as a function of deciduous broadleaf species (%) (proportion of deciduous broadleaf species) in the boreal forest. Proportion of deciduous broadleaf species affecting mean winter albedo in **(a)** post-harvest stands ($n = 17$) and **(b)** in post-fire stands ($n = 20$); mean spring albedo in **(c)** post-harvest stands ($n = 8$) and **(d)** post-fire stands ($n = 6$); mean summer albedo in **(e)** post-harvest stands ($n = 20$) and **(f)** post-fire stands ($n = 15$); and mean fall albedo in **(g)** post-harvest stands ($n = 8$) and **(h)** post-fire stands ($n = 7$). Note that albedo values of some 0–4-year-old stands were omitted from this analysis because these young sites had only a few seedlings of deciduous broadleaf species. If we included them here, the percentage of deciduous broadleaf species for these sites became 100 %, which was misleading compared to other sites; they were not zero either. Thus, the percentage of deciduous broadleaf species of these sites were excluded from this analysis and were considered as the ground vegetation cover (%) (Fig. 8). Additionally, percentage deciduous broadleaf species of some secondary-data sites were not reported, so were excluded from this analysis. The color scale (firebrick to dark green) indicates the range of stand age (young to mature), which is used to demonstrate the effect of stand age on seasonal albedo in the albedo-deciduous broadleaf species space.

0.01) (Fig. 5d). Proportion of deciduous broadleaf species was generally related to albedo via a simple exponential model ($\Delta$AIC $= -25.4$), and all estimated model parameters were significant ($p < 0.01$) (Fig. 6h). In the case of fall albedo in post-harvest stands, there is an apparent decline in nearly pure stands (Fig. 6g), with a better fit of the dou-

ble exponential model. We speculate that very dark post-senescence leaf litter of aspen may be the main cause for this effect.

We also fitted GLMs for albedo in post-fire and post-harvest stands as a function of stand age, proportion of deciduous broadleaf species, canopy height, and ground veg-

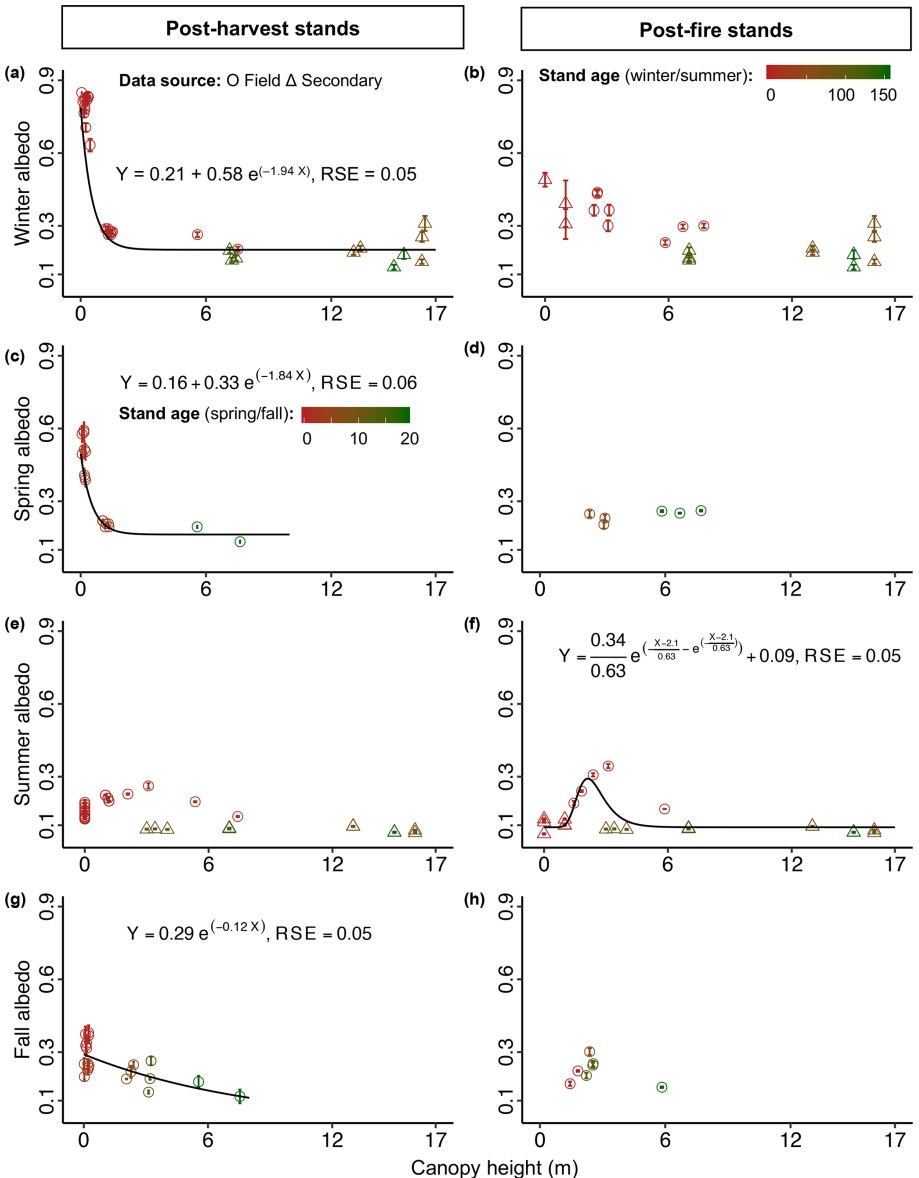

**Figure 7.** Mean seasonal albedo ($\pm$ SE) as a function of canopy height (m) in the boreal forest. Canopy height affecting **(a)** mean winter albedo in post-harvest stands ($n = 31$), **(c)** mean spring albedo in post-harvest stands ($n = 16$), **(f)** mean summer albedo in post-fire stands ($n = 23$), and **(g)** mean fall albedo in post-harvest stands ($n = 20$). In **(b)**, **(d)**, **(e)**, and **(h)** canopy height is not a significant predictor of the corresponding mean seasonal albedo; thus, no model is fitted to the data points. The color scale (firebrick to dark green) indicates the range of stand age (young to mature), which is used to demonstrate the effect of stand age on seasonal albedo in the albedo-canopy height space.

etation cover for all seasons after excluding the data from secondary sources. Results from this analysis indicate that best-fit models had the same structure (compared to the models with secondary data) and same variables were found to be significant predictors of seasonal albedo in post-fire and post-harvest stands (Table S1).

## 4 Discussion

Our results provide evidence for significant effects of disturbance type on the albedo of boreal forest systems, with post-harvest stands showing much higher albedo values in winter and spring months than post-fire stands. Stands of both disturbance types also showed strongly age-dependent patterns in albedo, with a transient peak in summer albedo at $\sim 10$ years; however, analyses also suggest that later post-disturbance changes are more gradual than anticipated, with

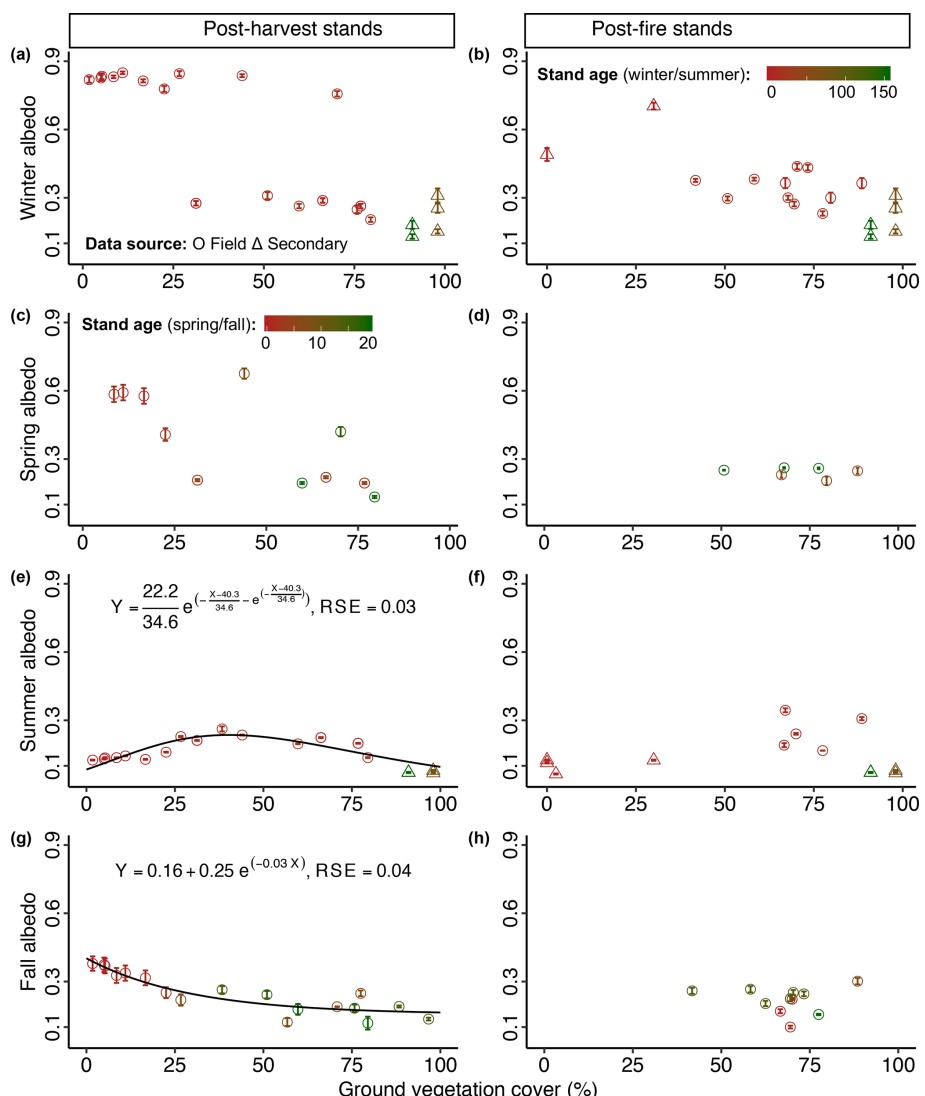

**Figure 8.** Mean seasonal albedo ($\pm$ SE) as a function of ground vegetation cover (%) in the boreal forest. Ground vegetation cover affecting **(e)** mean summer albedo in post-harvest stands ($n = 22$) and **(g)** mean fall albedo in post-harvest stands ($n = 18$). In **(a)**–**(d)**, **(f)**, and **(h)** ground vegetation cover is not a significant predictor of the corresponding mean seasonal albedo; thus, no model is fitted to the data points. The color scale (firebrick to dark green) indicates the range of stand age (young to mature), which is used to demonstrate the effect of stand age on seasonal albedo in the albedo-ground vegetation cover space.

dynamics continuing for decades following stand closure. The proportion of deciduous species also had large effects on stand albedo – generally larger than stand-age effects as indicated by overall lower residual standard errors of deciduous broadleaf species (%) regression models (Figs. 4–5 vs. 6) – and showing a positive response in all seasons and for both disturbance types.

## 4.1 Albedo in post-harvest and post-fire stands

Mean albedo in post-harvest stands was significantly higher than in post-fire stands in winter and spring, marginally higher in summer, and similar in fall (Fig. 2). A similar pattern in albedo differences was also observed when the stand-age effects on albedo were statistically controlled. The magnitude of differences in winter and spring values (0.22 and 0.08, or 63 % and 34 % increases relative to post-fire values) is large – comparable to albedo differences observed between biomes (Stephens et al., 2015). During snow-covered seasons (winter and spring), charcoal residues in post-fire stands are usually covered with snow, and thus stand structure and composition act as dominant drivers of albedo (Lyons et al., 2008; Amiro et al., 2006b; Liu et al., 2005). Deciduous broadleaf species made up 37.8 % of basal area in post-fire stands and 55.4 % in post-harvest stands: the higher percentage of dark conifer leaves is expected to result in lower winter/CE8 spring

**Biogeosciences, 16, 1–18, 2019**                                    **www.biogeosciences.net/16/1/2019/**

albedos in post-fire stands compared to post-harvest stands (Betts and Ball, 1997). However, immediately after a stand-replacing fire, the presence of black carbon (charcoal and soot) in the snow can reduce early winter albedo and possibly enhance spring snowmelt by absorbing solar radiation (Qian et al., 2009; Conway et al., 1996). During late spring when snow cover is shallow, it is also likely that charred branches and stems protrude through the snow and reduce albedo. Additionally, by the time of snowmelt, snow generally has accumulated particulate matter and has lower albedo compared to fresh snow (Conway et al., 1996). During this time of the year, latent heat flux from the melting snow is usually very high. Thus, from an energy balance perspective, it is important to note that albedo differences in late spring may be less important as turbulent and latent fluxes likely dominate (Conway et al., 1996).

In snow-free seasons (summer and fall) the marginal differences in mean albedo between post-harvest and post-fire stands can partly be attributed to recovery of ground vegetation in post-fire stands (0–5 years old) compared to post-harvest stands (Bartels et al., 2016), and to the vegetation covering dark charcoals in older (> 5 year) post-fire stands (Randerson et al., 2006). Soon after a fire, the presence of early-successional plants (Johnstone et al., 2010) can increase surface albedo of post-fire stands because of their higher albedo relative to charcoal (Amiro et al., 2006b; Betts and Ball, 1997). This effect is expected to offset the albedo difference between post-harvest and post-fire stands. In the first year following disturbance events, we might expect lower snow-free albedo in post-fire stands than in post-harvest stands because of high charcoal occurrence on the soil surface (Lyons et al., 2008; Chambers and Chapin, 2002). However, our soil reflectance data indicate that soils from 4-year-old post-fire stands unexpectedly showed significantly higher reflectance in the visible spectrum than did post-harvest stands (with the pattern reversed in the NIR spectrum) (Fig. 3a). Similar patterns in spectral response were recently observed in a biochar-amended agricultural soil relative to the control (Zhang et al., 2013). Soils from older post-harvest stands (11 and 19 years old), as expected, showed higher reflectance in the visible and NIR spectra compared to post-fire stands of similar age (Fig. 3b). Most post-harvest stands exhibited patches of charcoal in surface soils, presumably originating from historical fires. "Legacy" charcoals have similarly been visually observed on the forest floor and within upper mineral soils even after a hundred years following wildfire in Scandinavian (Ohlson et al., 2009) and Russian (Wallenius, 2002) boreal forests. The importance of legacy soil charcoal on surface albedo of harvested stands has not been considered previously to our knowledge. Charcoal reflectance is highly dependent on charring conditions (e.g., temperature, oxygen content) (Hudspith et al., 2015) and may possibly change with weathering; these processes require additional study in the context of albedo and surface energy balance.

Although charcoal residues likely have some influence on post-disturbance albedo, our results from both snow-covered and snow-free seasons strongly suggest that fire residues on the ground cannot explain the observed differences in albedo between post-harvest and post-fire stands. This result is consistent with the generalization that stand structure and composition are the main drivers of surface albedo and energy balance in the boreal forest (Amiro et al., 2006a).

## 4.2 Albedo convergence with stand age in post-harvest and post-fire stands

Compiled data for winter and summer albedos from post-harvest and post-fire stands indicate that changes in surface albedo continue for some decades following disturbance (Fig. 4a–d). This finding does not support our second hypothesis that albedo shows an early saturation near the onset of the "stem exclusion" phase. The rationale behind this hypothesis was that high productivity of mixedwood stands would result in more rapid canopy closure and attainment of peak LAI. Studies using remote sensing techniques, mostly focused on single-species stands, suggest that albedo in both post-harvest and post-fire stands commonly saturates at ∼ 40–80 years after harvest or fire (Bright et al., 2015b; Kuusinen et al., 2014; Lyons et al., 2008; McMillan and Goulden, 2008), consistent with our findings. Our results also suggest that gradual changes in species composition through later stages of succession are an important driver of stand albedo. Stand structural features such as canopy height (in winter) and ground vegetation cover (in summer) usually increase with stand age (Bartels et al., 2016) and might additionally contribute to a gradual reduction in albedo (Hovi et al., 2016) after ∼ 25 years (Table 2, Figs. 7 and 8).

The shape of best-fit curves for winter albedo vs. stand age (exponential decay) of post-harvest (Fig. 4a) and post-fire (Fig. 4b) stands are similar to other studies (Bright et al., 2015a; Kuusinen et al., 2014; Lyons et al., 2008; McMillan and Goulden, 2008; Amiro et al., 2006b); however our results diverge markedly for summer albedo. Our best-fit curves for summer albedo vs. stand age for both post-harvest and post-fire stands showed pronounced peaks in early albedo described by double exponential functions (Fig. 4c–d), whereas Amiro et al. (2006b) described data with a negative linear relationship, and other remote sensing-based studies have used exponential decay curves (e.g., Kuusinen et al., 2014). In contrast, Lyons et al. (2008) and Randerson et al. (2006) found summer albedo of post-fire stands were related to stand age via a humped-shape curve, and albedo reached its **CE9** peak at ∼ 20 years and gradually leveled off at ∼ 50 years after fire, which closely corresponds to our findings (although our observed peak is at ∼ 10 years post-disturbance; Fig. 4c–d). We suggest that most prior studies with sparser or more noisy data sets may have missed this early peak pattern. Immediately after fires and harvesting (because of high soil moisture, decaying CWD, legacy charcoal etc.), the

summer albedo of post-harvest and post-fire stands is expected to show a low value (also see Sect. 4.1 and Fig. 3), which sharply increases as dark ground is covered with early successional pioneer species (Lyons et al., 2008; Randerson et al., 2006; Amiro et al., 2006b; Betts and Ball, 1997). This sharp increase continues until $\sim$ 10–20 years of stand age but then decreases slowly until $\sim$ 50 years and saturates – consistent with the patterns found in other seasons.

We did not have albedo data from late-seral stands for spring and fall seasons. In post-harvest (Fig. 5a) and post-fire (Fig. 5b) stands, spring albedo values did not show strong patterns with stand age, and the patterns were disturbance specific (exponential decrease vs. negative exponential growth, respectively). Results from Kuusinen et al. (2014), Lyons et al. (2008), and Randerson et al. (2006) also suggest that patterns of spring albedo as a function of stand age can be disturbance specific. In post-harvest stands, Kuusinen et al. (2014) found that spring albedo was high immediately after harvest, decreased exponentially until $\sim$ 50 years and then saturated. However, in post-fire stands, Lyons et al. (2008) and Randerson et al. (2006) found hump-shaped patterns with a peak at $\sim$ 10–15 years, and subsequent declines, similar to the winter albedo pattern. As discussed in Sect. 4.1, disturbance-specific responses may partially be attributed to the presence of black carbon (charcoal or CE10 soot) on snow immediately after a fire, which can substantially reduce snow albedo (Qian et al., 2009). Trends in fall albedo values with stand age in post-harvest (Fig. 5c) and post-fire (Fig. 5d) stands showed stronger patterns than spring but showed CE11 similar disturbance-specific responses. Immediately after harvest, fall albedo was high and exponentially decreased as stand age increased. Increased fall albedo in recent post-harvest stands may be due to contributions to senescing leaves and to snow in the late fall (Amiro et al., 2006b; Liu et al., 2005). In contrast, fall albedo immediately after fire was low (possibly because of charcoal or soot residues as discussed above) and increased with stand age.

It is important to note that stand age itself is not a biophysical driver of seasonal albedo in post-disturbance stands; instead, it acts a proxy for multiple drivers, including commonly measured stand structural characteristics but also less commonly measured features that influence albedo. As can be seen from Figs. S2–S5, stand age in the mixedwood boreal forest is related to stand structural attributes such as canopy height, ground vegetation cover, and proportion of deciduous broadleaf species. However, our modeling results indicate effects of stand age on seasonal albedo that are independent of these measured variables, suggesting the importance of other, non-measured features or processes correlated with stand age. For example, the abundance and exposure of charcoal in the soil is not a commonly considered stand structural feature, but we found that it can affect stand albedo substantially both in post-harvest and post-fire stands. In the years following disturbance, increasing vegetation cover and leaf litter deposition are expected to reduce charcoal effects

on albedo. Additional processes and structures likely related to stand age but difficult to measure in situ include coarse and fine woody debris that influences surface roughness and snow exposure, and soil moisture that strongly influences bare soil albedo in snow-free conditions.

The ability to empirically predict forest surface albedo from stand age using the models presented in this study may specifically be useful to the forest managers to develop climate-sensitive forestry practice. Predicting albedo has been a long-standing problem in climate simulations (Bright et al., 2015b; Kuusinen et al., 2012; Qu and Hall, 2007). Our findings indicate that there are important qualitative differences in the post-disturbance albedo patterns between seasons in boreal forests. These differences need to be considered in enhancing albedo predictability of land surface models.

### 4.3 Deciduous broadleaf species as a key determinant of surface albedo in the post-harvest and post-fire stands

Our results indicate that the proportion of deciduous broadleaf species is a strong predictor of albedo irrespective of disturbance type and in most cases is a better predictor than stand age (Figs. 4–6). Using remote sensing techniques, Kuusinen et al. (2014) also found that stand age alone was not consistently the best predictor of stand albedo in the boreal forest. We found a similar mean model residual sum of errors for snow-covered seasons and snow-free seasons (Figs. 4–5 vs. Fig. 6), indicating that the proportion of deciduous broadleaf species is similarly important in both cases. These findings strongly support our third hypothesis that stands with a higher proportion of deciduous broadleaf species show higher albedo than conifer-dominated stands, but also that this effect is pronounced under both snow-covered and snow-free conditions. Except for fall CE12 post-fire stands, the relationship between albedo and proportion of deciduous broadleaf species approximated by an exponential saturating curve in which albedo declined rapidly where the proportion of deciduous broadleaf species fell below 25 %–50 %. Fall albedo in post-fire stands, on the other hand, was found to be even more sensitive, with a drop in fall albedo at a proportion of deciduous broadleaf species below 80 %. We speculate that this sensitivity was related to exposure of fire residues in early stand development.

Overall our results indicate a strong dependency of seasonal albedo on the proportion of deciduous broadleaf species both in post-harvest and post-fire stands. This effect provides a strong link between albedo and successional patterns in mixedwood boreal forests. Prior studies addressing this relationship (e.g., Lyons et al., 2008; Amiro et al., 2006b) have suggested that increasing deciduous tree cover results in increased albedo values from stand initiation to $\sim$ 25 years of stand age; thereafter, conifers start dominating the canopy, canopy height increases, and albedo decreases gradually un-

til $\sim 50$ years of stand age before reaching a steady state. The data presented in the current study provide a somewhat different picture of these trends, in that patterns show important quantitative differences depending on the season and disturbance type. The importance of deciduous broadleaf species in the albedo signal over $\sim 50$ years of stand development suggests that slow successional changes in species composition are a main driver of the age-related patterns in mixedwood boreal forests albedo in later successional stages. The dynamics of this pattern is likely to depend on the intensity and frequency of disturbance, edaphic conditions, species abundance, and climate (Taylor and Chen, 2011). For example, in dry nutrient-poor boreal stands, deciduous broadleaf species-driven albedo might never occur, as such stands are commonly dominated by jack pine (Taylor and Chen, 2011); however, in mesic moderate-nutrient-rich stands, deciduous broadleaf species can dominate for $\sim 100$ years (Cogbill, 1985). Future studies should prioritize robust modeling of boreal succession pathways under different biotic or abiotic conditions to properly characterize stand albedo.

## 5 Conclusions

This study presents the first available data on albedo patterns in boreal forests for all four seasons as well as the first comparisons of albedo in post-fire and post-harvest stands in the mixedwood boreal forest. The new data presented here are from 15 instrumented sites each monitored for 4 years, providing 60 site-years of measurements, all in mixedwood boreal forests, which are the most CE13 important forests from a forest management perspective, for which there are almost no prior ground-based albedo measurements. Analyses of this unique dataset indicate the following: (i) winter and spring albedo values are substantially higher in post-harvest than in post-fire stands; (ii) post-disturbance patterns of albedo recovery in boreal mixedwood stands are strongly influenced by changes in species composition; (iii) there are important stand-age-related dynamics in albedo in the first 20 years following disturbance events that have been missed by prior studies.

These findings have important implications for climate-friendly forest management practices. Since the proportion of deciduous broadleaf species is a strong predictor of seasonal albedo, stand-level albedo can be increased by enhancing the proportion of deciduous broadleaf species in a stand. Precisely this approach has recently been suggested as an adaptation and mitigation strategy to counter negative climate forcings of boreal forest (Astrup et al., 2018), but empirical data from actual managed stands have been lacking. Historically, forest managers have commonly sought to decrease or eliminate deciduous species and enhance conifers. However, there is strong evidence that local tree diversity enhances productivity in boreal forests as in other systems (Paquette and Messier, 2011)) and in particular that mixedwood

boreal forests, including both conifers and deciduous trees, show high productivity (MacPherson et al., 2001; Zhang et al., 2012). Management to increase the proportion of deciduous broadleaf species in managed boreal forests (for example, simply by avoiding chemical herbicide used to kill deciduous broadleaf species or retaining deciduous broadleaf species seed trees) could thus be a win-win scenario for enhanced carbon sequestration via primary productivity, and climate mitigation via enhanced albedo.

In climate modeling studies, albedo estimation for boreal forests have commonly been achieved by highly simplified representations of vegetation dynamics (Thackeray et al., 2019). In a recent study, Bright et al. (2018) pointed out that overlooking stand structural and compositional properties over the successional trajectory is likely to substantially bias radiative forcing estimates in the boreal forest. Ground-based estimates such as those presented are essential: at high latitudes when solar zenith angle is high ($> 70°$), satellites such as MODIS often provide poor-quality albedo data due to spatial heterogeneity of the landscape pixel signature and performance degradation of atmospheric correction algorithms (Bright et al., 2015a; Wang et al., 2012). Our findings based on field data are thus important in evaluating and potentially improving albedo predictions in land surface characterizations with climate models and in improving albedo estimates derived from remote sensing. In addition, our results point to the importance of slow ecological succession as a driver of age-related patterns in albedo, suggesting that future models should explicitly incorporate these ecological processes to better predict long-term trends in climate forcings in boreal forests.

*Code availability.* R (version 3.5.1) codes used in the data analysis can be requested from CE14 Mohammad Abdul Halim (abdul.halim@mail.utoronto.ca).

*Data availability.* Data used to produce the graphs are also available by request from the CE15 corresponding author (abdul.halim@mail.utoronto.ca).

*Supplement.* The supplement related to this article is available online at: https://doi.org/10.5194/bg-16-1-2019-supplement. TS3

*Author contributions.* MAH, HYHC, and SCT designed the experiment. MAH analyzed data with inputs from SCT and HYHC. MAH wrote the paper with edits and comments from HYHC and SCT.

*Competing interests.* The authors declare that they have no conflict of interest.

*Acknowledgements.* Authors acknowledge the assistance from Jillian Bieser, Kyle Gaynor, Lutchmee Sujeeun, Jack Richard, Jad Murtada, Anna Almero, and Hiro Sato during experiment setup and data collection. We thank Mark Horsburgh for his helpful comments on an earlier version of this paper CE16 and Shannon Brown for her help during the field comparison of pyranometers. We would also like to thank two anonymous reviewers and the editor for their critical comments, which helped to improve the paper. This study was funded by the NSERC (Natural Sciences and Engineering Research Council of Canada) Strategic Grant (grant number: STPGP 428641) and the NSERC Discovery Grant (grant number: RGPIN 06209).

*Financial support.* This research has been supported by the NSERC (Natural Sciences and Engineering Research Council of Canada) Strategic Grant (grant no. STPGP 428641) and the NSERC Discovery Grant (grant no. RGPIN 06209). TS4

*Review statement.* This paper was edited by Sebastiaan Luyssaert and reviewed by two anonymous referees.

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

## Remarks from the language copy-editor

CE1 Please note the slight edits to the affiliations section.

CE2 Do you mean here "regional or global mean surface temperatures"? The use of a forward slash is grammatically ambiguous and can mean "and", "or", or "and/or".

CE3 Can these be considered as ratios?

CE4 Please check that the meaning of your sentence is intact.

CE5 Please define AIC.

CE6 Do you mean measurements or samples?

CE7 Please check that the meaning of your sentence is intact.

CE8 Do you mean here "winter and spring"?

CE9 Please confirm the change.

CE10 Please confirm the change.

CE11 Please confirm the change.

CE12 Please check and rephrase this sentence as it is not entirely clear what is meant. There appears to be some words missing.

CE13 Please check that the meaning of your sentence is intact.

CE14 Please note the slight edit.

CE15 Please note the slight edits.

CE16 Please note the slight edits to this section.

## Remarks from the typesetter

TS1 The composition of Figs. 1 and 3–8 has been adjusted to our standards.

TS2 Please note that units have been changed to exponential format. Please check all instances.

TS3 Please send a new supplement as a \*.pdf without the title, authors, correspondence author, etc. as we will generate a supplement title page during publication (with a citation including the DOI), which will contain this information.

TS4 Please note that the funding information has been added to this paper. Please check if it is correct. Please also double-check your acknowledgements to see whether repeated information can be removed or changed accordingly. Thanks.

TS5 Please provide date of last access.

TS6 Please check URL and provide date of last access.

TS7 Please provide the page range if there is more than one page.

TS8 Please provide the full last names for all authors.

TS9 Please provide the volume.

TS10 Please provide the page range or article number.

TS11 Please provide the page range or article number.

TS12 Please provide the page range or article number.

TS13 Please provide date of last access.

TS14 Please provide the page range or article number.

TS15 Please provide date of last access.

TS16 Please provide the page range or article number.

TS17 Please provide place of publication.

TS18 Please provide date of last access.

TS19 Please provide the page range or article number.

TS20 Please check URL and provide date of last access.

TS21 Please provide date of last access.

TS22 Please provide date of last access.