# Peer review of "Stand age and species composition effects on surface albedo in a mixedwood boreal forest"

_Biogeosciences, 2018_

## Referee Comment (RC1) · Anonymous Referee #1 · 27 Jan 2019

General comments

This study investigates the effect of disturbance type (harvest, fire) on boreal forest albedo, based on in situ albedo measurements. The authors conclude that i) post-harvest albedo is higher than post-fire albedo, ii) albedo saturates at ∼50 years' age after both disturbance types (which is later than the authors expected: they expected saturation at ∼25 years' age or earlier), and iii) successional changes in species composition are a key driver of age-related patterns in albedo. I see high risk that the conclusions are not valid because:

1. The authors complement their data with data taken from previously published papers (which they call 'secondary sources'). When looking at those papers in detail it is noticed that they used pyranometers working at full range of the solar spectrum (from approx. 300 to 2800 nm), while the data collected by the authors is recorded at visible and near infrared spectral regions (300–1100 nm). This can result in substantial differences between the data sets. Another problem is that the effects that are seen may be due to differences in climate (particularly snow depth, snow properties, and extent of snow-covered period), rather than forest structure or species proportions. The most distant study site (Alaska) is thousands of kilometres away and located in different latitudes than the main study site ($\sim$65°N compared to 49.55°N). Therefore, it is likely that not only climate, but also the forest structure (e.g. height, canopy closure) as function of age, differs notably from the forests measured by the authors.

2. The amount of data is relatively small (only 15 plots + those obtained from secondary sources). For example, the conclusion that wintertime post-harvest albedo is higher than wintertime post-fire albedo seems to be due to some post-harvest plots showing very high albedos. The variation in albedo among post-harvest plots is large (Figure 4a). Removing some of the post-harvest plots with high albedo would probably result in notably different conclusions. Another example is the conclusion that albedo saturates later than at 25 years' age (but at no later than 50 years' age). I think that it is impossible to make such conclusion, because there is no data measured for stands aged 20–50 years.

3. The error sources and quality control of the measurements are not described in detail enough. The highest summer- and wintertime forest albedos measured by the authors are towards the higher end of what has been reported earlier. This may be because of limited spectral range in the measurements (i.e. not a measurement error), but it is difficult to say for sure, because this is not discussed by the authors.

From 1 it follows that the secondary data sources are not comparable with the authors' measurements and should be left out. I appreciate value of in situ data and the effort that the authors have put in the experiment. It might be possible that the authors'

measurements alone would result in interesting conclusions. However, this is difficult to evaluate based on the data and figures presented. I provide specific comments below to help the authors improve their work.

Specific comments

L17-18: It would be useful to state which stand ages these values (63%, 24%) apply.

L21: It is not clear what is meant by "seasonal albedo". I suggest defining the concept before its usage.

L21-23: I agree that change in species composition when the forest gets older is one driving factor of albedo changes. However, also the forest structure changes when the forest gets older. For example, increasing canopy closure reduces the visibility of ground surface, and increasing tree height/canopy closure increase the shadow fraction. These both reduce forest albedo. There is lots of empirical evidence (at least based on satellite albedo measurements) in literature suggesting that albedo of coniferous forest changes with stand age, even though the species composition does not change. Thus, I think that your statement here is too strong. Species composition is one driver of age-related albedo changes, but based on the data presented, I would not say that it is a "key driver".

L34: It might be useful to clarify what is meant by "the relative stability of the atmospheric temperature profile".

L70-72: I doubt that the legacy charcoal from fires that happened several years or decades ago would influence albedo at the time of harvest.

L73-75: This sentence is also a bit unclear. It needs more clarification how (through which physical mechanisms?) decomposition processes and plant colonization would influence albedo.

L96-97: The first hypothesis is a bit contradictory to what is stated on L65-68 (that post-harvest stands typically have higher proportion of broadleaved species). Large

proportion of broadleaved species should lead to higher albedo.

L105: I think it is important to explain (in qualitative, descriptive terms) what kind of structure do the post-fire stands have. Do they have lots of standing dead trees? How severe were the fires?

L107: It is not clear whether all three replicate plots were in separate stands or in the same stand?

L111-119: Would it be possible to show the forest structural variables for each study plot in a table? This way the reader would get better understanding of the forest structure and species composition and their development through stand age.

L117: What does the abbreviation LFH stand for?

L121-124: How was the placement of the albedo measurement towers? I guess that the stands were surrounded by older (higher) forests? How far from the stand edges the towers were placed? Did the surrounding forests block a portion of the incoming diffuse radiation and is there a possibility that this would have affected the measurements?

L122: Due to limited spectral range (300–1100 nm) the upper end of solar spectrum (from 1100 nm up to 4000 nm) is left out and therefore the measured albedo is not full shortwave albedo. I looked at the methods of the papers providing secondary data sources, and noticed that they used full solar spectrum: -Chambers and Chapin (2002), Liu et al. (2005): Eppley precision spectral pyranometer, 285–2800 nm -Amiro et al. (2006a): Kipp and Zonen CNR1, 305 to 2800 nm -Amiro et al. (2006b): Kipp and Zonen CM3, 305 to 2800 nm

L126-127: What kind of quality control procedures were applied in the data processing? The explanation in the manuscript gives an impression that data from all days were useful and no outliers etc. needed to be removed. Did you remove some observations/days due to low quality?

L131: "deciduous broadleaved area". Does this refer to basal area, or something else?

L132-133: The diameter limit is much stricter than the height limit. Usually trees with 5 cm diameter at breast height have height at least ~5 meters. This means that you do not need the height limit at all, because the diameter limit already excludes all trees with height less than 5 m. The diameter limit is also quite high considering the young age of the forests. I do not know exactly how fast the trees grow in the study area, but I would guess that forests with ages from 0 to 10 years have only few trees (if any) with diameter at breast height exceeding 5 cm. Is the high diameter limit the reason why some stands are missing in Figure 6 (that shows the albedo dependence on broadleaved proportion)?

L137-140: Which of the measurement years (2013-2017) were used in calculation of the age? Or did you treat each year as separate observation and thus the age differed depending on which year was used?

L143-153. How does the climate in the secondary sites (particularly snow depth, snow properties, extent of snow-covered period) differ from the site in Ontario? This is very important because if the climate differs markedly, then the observed differences between post-fire and post-harvest stands are not solely due to stand structure and species composition.

L165: "top-surface-specular-included (diffuse and direct) reflectance" is a bit awkward definition. If the reflectance values were measured with integrating sphere setup (collimated light used for illumination, and the reflected radiation collected over the hemisphere), then I would express it something like: "directional-hemispherical reflectance factor of the top-surface of the soil sample".

L169: Does "ten different locations" refer to ten different locations within the sample? How is a location defined (how large area is covered by one measurement)?

L169: What does "Boxcar width 5" mean? It needs more explanation.

L187-196: Are the selected model shapes based on physical nature of the phenomena

studied, or are they just chosen to give the best model fit to the observed data? For example, in Figure 6g it is difficult to imagine a physical reason why albedo would first increase as function of broadleaved percentage, and then decrease again as the broadleaved percentage approaches 100%.

L209-210: Variability of albedo in post-harvest stands in winter and springtime is indeed very high. For example, in Figure 4a it is seen that the albedos of the young forests (<50 years) vary from approx. 0.2 to almost 0.9. I think that 0.9 is very rarely observed except for pure snow surfaces with no vegetation. Are you sure that the variation is not caused by measurement errors? This is why I suggest reporting the details of your measurement and quality control procedure in detail. Another thing that caught my attention is the high albedo (approx. 0.3) of some stands in summertime (Figure 4c,d). Albedo values 0.2 (at solar noon) are rarely reported in boreal forests. Your values are approx. 50% higher than that. Is this because of the limited spectral range of the pyranometer?

Figures 2, 4–8: The number of observations varies between figures. It is not clear what constitutes an observation? There were 15 plots but the number of observations can be much higher than this. Is each year treated as a separate observation? Text on line 128 suggests this. If each year is treated as a separate observation, how does it affect/violate assumptions (independence of observations) in the statistical models?

Section 3.2: It might be possible to weight the observed ground spectra with incoming solar radiation, to calculate albedos of the ground surface. This way, it would be easier to link age-related changes in ground albedo to age-related changes in forest albedo.

Figure 6: Why Figure 6 does not contain all data presented in Figure 4 (high winter albedo values close to 0.9 are not presented in Figure 6)?

L254-255: Earlier (in Section 2.5) you state that only fall albedos were modelled with double-exponential model, but here you say that also summer albedo was modelled with double-exponential model. Which one is true?

L275: Please explain in more detail what is meant by "to avoid modelling complexities". Does it mean that the model did not converge if it was too complex?

L287-288: I think word "dramatic" is too strong.

L290: I doubt that the conclusion that albedo saturates at 50 (rather than at ∼25) years' age is valid. You do not have any measurement data between 20-50 years (Figure 4).

L291: Please explain in more detail how you determine that the effect of broadleaved proportion is larger than the stand age effect?

L306-308: I think this sentence is a bit misleading. I would expect that from an energy balance perspective late spring is more important than winter, because the amount of incoming solar radiation is larger in late spring.

Technical corrections

L177-178: Near-infrared range must have a typo [700–100 nm].

L206-207: "post-harvest" repeated two times.

Table1: It is a bit difficult to see the difference between italic vs. regular font. Perhaps bold vs. regular would be a better choice?

L357: "albedo" repeated two times.

---

## Referee Comment (RC2) · Anonymous Referee #2 · 29 Jan 2019

By combining literature data with novel in-situ measurements and via chronosequencing, the study by Halim et al. analyzes temporal trajectories in surface albedo following harvest and fire disturbances in southern boreal mixedwood forests. The main findings are that i): winter and spring surface albedos following harvest disturbances are higher than those following fire disturbance; that ii) both winter and summer surface albedos "saturate" at around 50 years, and that iii) successional changes in species composition are a key driver of post-disturbance albedo dynamics.

I have several major concerns about the study methods that call to question these findings. My first and largest concern surrounds the extensive use of albedo data sourced

from the literature (referred to as "secondary data") which are connected to sites located hundreds to thousands of kilometers away. Although the dominant species compositions across sites may be similar, stand structure and other important site-specific attributes affecting the surface albedo may differ greatly across sites. These include differences in geology and soils (affecting albedo via their controls over understory vegetation compositions, soil moisture retention, growth rates), differences in latitude (affecting the direct albedo component via differences in solar geometry), and – most importantly – differences in local climate (affecting albedo via controls over soil moisture, vegetation growth and phenology, length of snow season, and important snow physical attributes such as snow depths, snow age, snow water contents). Without controlling for differences in these important site-specific factors it is difficult to arrive at robust conclusions regarding albedo-age dynamics, albedo-species composition dynamics, albedo-canopy height dynamics, and albedo-ground cover dynamics. Regarding the albedo-age dynamic, for instance, asymptotes of the presented exponential models in Figure 4 seem to be heavily influenced by the "secondary" data comprising all data points beyond 19 years. Regarding the albedo-species composition dynamic, the "secondary" data points in Figure 6 for "Summer" and "Winter" seem to be heavily influencing the y-intercepts and thus affecting the model functional form and shape parameters. Secondary data points in Figure 7 also appear to heavily influence the model fits (or lack thereof) for the "Summer" albedo-canopy height dynamics.

A second methodological concern which is also related to the augmentation of the in-situ sample with literature ("secondary") data is the difference in the definition of albedo. Much of the secondary albedo data are for a broader spectral range (e.g., 295-2800 nm) than what is measured in-situ Âň at the authors' own study sites (i.e., 300-1100 nm). This is important given the high albedo of vegetation in spectra above 1000 nm and given the sensitivity of the shortwave near-infrared broad band (1300-2500 nm) to differences in boreal tree species (see Hovi et al. 2017 → https://doi.org/10.14214/sf.7753 ).

I also have some concern about the study's scientific value, irrespective of my concerns about the methods. None of the three major findings listed above are novel and can be distilled from a diligent review of the boreal forest albedo literature (e.g., post-fire: Lyons et al. 2008; Randerson et al. 2006; Amiro et al. 2006b; Liu et al. 2005; Wang et al. 2016 → https://www.sciencedirect.com/science/article/pii/S0034425716300888 post-harvest: Kuusinen et al. 2016; Kuusinen et al. 2014; Bright et al. 2013; Hu et al. 2018 → https://agupubs.onlinelibrary.wiley.com/doi/full/10.1029/2018MS001403 ).

Further, the study is motivated by the need to "improve climate model parameterizations" but the authors have made no attempt to explain how their results can/will achieve this. How will the presented statistical functions or empirical insights be applied in a climate modeling context, either for improving existing parameterizations in a climate model directly or for use as a climate model benchmarking/evaluation tool? Albedo parameterizations in most climate models are process-oriented and intimately tied to important forest structural attributes like leaf area index which the authors have not included. Model parameterizations are also largely oriented around important local meteorological state variables (i.e., near surface air temperatures, wind speeds, precipitation type and frequency, snow depth, etc.) which are absent in the paper. This makes it difficult to discern the conditions under which the reported findings may be applied to evaluate climate model predictions. Further, since the reported albedo dynamics for the post-harvest case are intimately connected to the specific management practices of the study region, without providing any detail about the prevailing management regime(s) of the study region it will be difficult for modelers to assess accuracy of simulated post-harvest albedo dynamics. As for the post-fire case, the finding that the near-term (< 25 yr) increases in summertime albedo are connected to pioneer birch succession (a finding reported in several of the references listed above) implies that any "improvement" to the albedo prediction capability of a climate model would need to target the vegetation dynamics routines of the model and not necessarily the "albedo parameterization" itself.

Given my concerns about the study's methods and low scientific significance, I find it difficult to recommend publication in BG. I also find it difficult to encourage a major revision involving a new analysis that excludes the use of "secondary" data given the limited number of field plots and given the narrow spectral band of albedo data that has been measured at those plots.

---

## Editor Comment (EC1) · Luyssaert (Editor) · 1 Feb 2019

Halim and co-authors present an interesting analysis on the age and species effect on albedo.

In the introduction previous reports are cited to list possible drivers. Stand age is, however, not among the listed drivers. In the discussion the authors do a reasonable good job in focusing the discussion on the physical drivers (fraction of deciduous trees, charcoal, stand structure, . . .). From this perspective it is surprising that the results section uses stand age as one of the independent variables to explain the changes in albedo (as reflected in the statistical models and the table). In my opinion, the authors

should better explain that the analysis with age is simply to describe the temporal evolution but that the additional analysis are intended to explain the physical drivers of these age trends. If this indeed reflects the thinking of the authors, the paper should be edited towards this message, e.g. no models should be fitted against age and several sentences throughout the manuscript should be rephrased. Nevertheless, if the authors interpret their results as an indication that age itself is a physical driver of albedo, it should be discussed how stand age (rather than structure) affects albedo.

The importance of this study for climate modelling should be rewritten in line with the state of art of albedo modelling through canopy radiative transfer models and the simplified canopy radiative transfer schemes that are used in the land surface schemes of climate models. The authors seems not be aware of recent work (Naudts et al 2016, Luyssaert et al 2018) that does account for the effect of stand structure, tree species, and forest management on albedo and the climate (including not only albedo but also transpiration and roughness). The impact on modelling efforts of the albedo observations presented in this study is largely overstated. Canopy radiative transfer schemes combine scattering parameters and simulated canopy structures to simulate the albedo. The albedo values reported in this study can be used to evaluate existing models but are unlikely to be useful to improve existing models as claimed in the text. It may be best to delete all references to model developments and focus the discussion and conclusions on the underlying processes and the remaining unknowns.

---

## Author Comment (AC1) · 14 Feb 2019

Response to Referee #1 We would like to thank Referee #1 for their time to read the manuscript, and to make thoughtful and critical comments, which will help to improve our manuscript. In this short response, we are only addressing a few issues raised by Referee #1 and the rest will be addressed in detail in the revised manuscript.

Comparability of albedo measured in the field and from secondary sources: The two most common sensors used in solar radiation measurements are thermopile (used in secondary data, $\sim$300–2800 nm) and silicon-crystalline (used in this study, 300–1100 nm) pyranometers (Mubarak et al. 2017). Referee #1 has pointed out that

these pyranometers have different spectral measurement ranges and, thus, not strictly comparable. We employed silicon-crystalline pyranometers in the present study due to their substantially lower cost, lower power requirements, and better properties for shedding snow and rime; we estimate that instrumenting and maintaining the sites with thermopile pyranometers would have resulted in additional research expenses of >CAD$100,000.

We respectfully argue that even though the silicon pyranometers have narrow spectral measurement range, silicon pyranometers can measure irradiance within +/- 3% error relative to thermopile pyranometers within a solar zenith angle of +/- 60 deg (Mubarak et al. 2017 and Li-COR/Onset pyranometer specifications). The relatively good agreement is partly because most of the solar energy (within 300–2800 nm) is concentrated within 400–1100 nm, and partly because the daily variation (overestimation/underestimation) in measured irradiance is due to changes in solar zenith angle (and associated silicon sensor responsiveness), and is mostly canceled out if averaged over 24 hours. Myers (2010) concluded that silicon pyranometers, when compared against WRR (World Radiometric Reference) cavity radiometer, can offer acceptable accuracy (within ∼5%), and that the agreement increases if averaged over longer time period. In our study, we have reported field measurements (albedo) averaged over entire seasons (3-month periods).

Relative accuracy in albedo measurements between thermopile and silicon pyranometers can be different than in irradiance measurements. The relative measurement error can stem from differences in spectral measurement range, behavior of sensing materials used, and biases to due to snow or rime accumulation. However, studies directly comparing relative performances of silicon pyranometers to thermopile pyranometer in measuring albedo are scarce. In a wheat field experiment, Francois et al. (2002) suggested that silicon pyranometers tended to systematically overestimate summer albedo, with an average bias of ∼0.03, and a maximum of 0.08 (at LAI ∼3) compared to a thermopile pyranometer. They also found that at lower LAI (∼2) the maximum

albedo difference was reduced and found no bias for bare soil conditions. (We note that the Francois et al. (2002) study, however, compared different sensors in different fields – and so almost certainly over-estimated differences). In our field sites LAI ranged from 0.0–2.1 and was <1 in most sites. In another study, Inge (1968) reported that silicon pyranometers overestimated plant, sand, water, and rock albedo by $\sim$0.03, and suggested caution should be taken for measuring snow albedo as the overestimation can be as high as 0.09. Stroeve et al (2005) found that silicon pyranometers can overestimate albedo by 0.04–0.09. This overestimation is mainly during the snow-covered period because of high snow reflectivity in <1100 nm range. However, they also point out that silicon pyranometers have benefits over thermopile pyranometers in measuring albedo in cold environments owing to their resistance to rime frost error (small form factor, no dome, and fast response time). Stroeve et al. (2005) made some simple corrections to the up-facing silicon pyranometer measurements (no correction for down-facing pyranometers) and calculated albedo was used to assess accuracy of MODIS albedo product (MOD43) (620–2155 nm) along with other thermopile pyranometers (305–2800 nm / 200–3500 nm) measurements. We note that several published studies (for example, Winkler et al. 2010, and Gleason and Nolin, 2016) have previously used silicon pyranometers to measure albedo of snow-covered forest floor.

Overall, the literature suggests that albedo measured by silicon pyranometers does show some systematic overestimation compared to thermopile pyranometers. Now the question is, what is the allowable error limit to make these two sensor measurements comparable? The answer is subjective. Even class one pyranometers in a lab environment typically vary by +/- 2% with respect to WRR cavity radiometer (Myers, 2010), and in field environment by +/- 5-7% (Stroeve et al 2005). Low cost and low maintenance of silicon pyranometers offer the flexibility to increase measurement replication, which is very important for stand-level albedo estimation. Given the spatial/structural heterogeneity of a forest stand, a single thermopile pyranometer measurement may not offer a better estimate of the stand albedo than albedo measured at multiple locations using silicon pyranometers.

We would also like to stress that secondary albedo data were not used in any statistical comparisons (t-tests, ANOVA) in our paper, but only used in model development to show general trends in albedo change with stand age.

On secondary albedo data usage from the Alaska (USA) site, even though stand characteristics were similar to our study area, we agree that because of different latitudes, snow cover duration, snow depth, and snow properties may strongly affect seasonal stand albedo. We will exclude Alaska data (2 data points for summer and winter each) from regression models in our revision; we have revised the analyses, and this modification does not affect any conclusion of the paper.

In the meantime, we have set up four (two down-facing and two up-facing) silicon pyranometers alongside four thermopile pyranometers (two down-facing and two up-facing) at the same height in an Ontario site to estimate relative albedo errors during winter, when it varies the most, under varying snow and sky conditions. We will report this error estimate in the Methods section (with more details on data processing) in our revised manuscript.

Referee comment: "I agree that change is species composition when the forest gets older is one driving factor of albedo changes. However, also the forest structure changes when the forest gets older. For example, increasing canopy closure reduces the visibility of ground surface, and increasing tree height/canopy closure increase the show fraction. These both reduce forest albedo. There is lots of empirical evidence (at least based on satellite albedo measurements) in literature suggesting that albedo of coniferous forest changes with stand age, even though the species composition does not change. Thus, I think that your statement here is too strong. Species composition is one driver of age-related albedo changes, but based on the data presented, I would not say it is a key driver."

Authors' Response: We agree with the point that changes in stand structure with age importantly influence albedo – and also note that this has been a main emphasis in

a number of prior studies. However, prior boreal forest albedo studies have focused almost entirely on pure stands, such as upland jack pine or lowland black spruce. In these pure stands species composition does not change (>90% remains the same species) with stand age. In such a stand albedo variation with stand age will necessarily be due mainly to changes in stand structural properties. Our study, however, was conducted in a mixedwood boreal forest, where species composition characteristically changes with stand age. In such stands, our results suggest that species composition can dominate other age-related drivers of albedo. As noted in the manuscript, mixedwood boreal forests are generally the most productive boreal systems, and most forest management occurs in such systems.

Referee Comment L122: "Due to limited spectral range (300–1100 nm) the upper end of solar spectrum (from 1100 nm up to 4000 nm) is left out and therefore the measured albedo is not full shortwave albedo. I looked at the methods of the papers providing secondary data sources, and noticed that they used full solar spectrum: -Chambers and Chapin (2002), Liu et al. (2005): Eppley precision spectral pyranometer, 285–2800 nm -Amiro et al. (2006a): Kipp and Zonen CNR1, 305 to 2800 nm -Amiro et al. (2006b): Kipp and Zonen CM3, 305 to 2800 nm."

Authors' Response L122: Please look at Table 1 of Liu et al. (2005): they have used LI 200R silicon pyranometers (LI-COR, Inc.) (400–1100 nm) in addition to Eppley precision spectral pyranometers for their 3-, 15-, and 80-year sites. Their Methods section implies that they have used LI 200R data at least for the 15-year sites. This is an example of a study where researchers have previously used silicon and thermopile pyranometers in combination.

References:

Francois, C., Ottle, C., Olioso, A., Prevot, L., Bruguier, N., and Ducros, Y.: Conversion of 400–1100 nm vegetation albedo measurements into total shortwave broadband albedo using a canopy radiative transfer model. Agronomie 22: 611–618. DOI:

10.1051/agro:2002033, 2002.

Gleason, K. E. and Nolin, A. W.: Charred forests accelerate snow albedo decay: parameterizing the post-fire radiative forcing on snow for three years following fire. Hydrological Process DOI: 10.1002/hyp.10897, 2016.

Inge, D.: On the use of silicon cells in meteorological radiation studies. Journal of Applied Meteorology 7: 702–707, 1968. Mubarak, R., Hofmann, M., Riechelmann, S., Seckmeyer, G.: Comparison of modelled and measured tilted solar irradiance for photovoltaic applications. Energies 10, 1688. doi:10.3390/en10111688, 2017.

Myers, D. R.: Comparison of direct normal irradiance derived from silicon and thermopile global hemispherical radiation detectors. Conference Paper (NERL/CP-550-48698) presented at SPIE Optics and Photonics, San Diego, California, August 1-5, 2010.

Stroeve, J., Box, J. E., Gao, F., Liang, S., Nolin, A., Schaaf, C.: Accuracy assessment of the MODIS 16-day albedo product for snow: comparisons with Greenland in situ measurements. Remote Sensing of Environment 94: 46–60, 2005.

Winkler, R., Boon, S., Zimonick, B., and Baleshta, K.: Assessing the effects of post-pine beetle forest litter on snow albedo. Hydrological Processes DOI: 10.1002/hyp.7648, 2010.

---

## Author Comment (AC2) · 14 Feb 2019

We thank Referee #2 for their critical comments.

Referee Comment 1: "I have several major concerns about the study methods that call to question these findings. My first and largest concern surrounds the extensive use of albedo data sourced from the literature (referred to as "secondary data") which are connected to sites located hundreds to thousands of kilometers away. Although the dominant species com- positions across sites may be similar, stand structure and other important site-specific attributes affecting the surface albedo may differ greatly across sites. These include differences in geology and soils (affecting albedo via their controls over understory vegetation compositions, soil moisture retention, growth rates), differences in latitude (affecting the direct albedo component via differences in solar geometry), and – most importantly – differences in local climate (affecting albedo via controls over soil moisture, vegetation growth and phenology, length of snow season, and important snow physical attributes such as snow depths, snow age, snow water contents). Without controlling for differences in these important site-specific factors it is difficult to arrive at robust conclusions regarding albedo-age dynamics, albedo-species composition dynamics, albedo-canopy height dynamics, and albedo-ground cover dynamics. Regarding the albedo-age dynamic, for instance, asymptotes of the presented exponential models in Figure 4 seem to be heavily influenced by the "secondary" data comprising all data points beyond 19 years. Regarding the albedo-species composition dynamic, the "secondary" data points in Figure 6 for "Summer" and "Winter" seem to be heavily influencing the y-intercepts and thus affecting the model functional form and shape parameters. Secondary data points in Figure 7 also appear to heavily influence the model fits (or lack thereof) for the "Summer" albedo-canopy height dynamics."

Authors' Response 1: On the use of secondary albedo data Referee #1 has pointed out similar issues. For the sake of brevity, we are not repeating the same response here. Please see the "Comparability of albedo measured in the field and from secondary sources" section of the Response to Referee #1.

Referee Comment 2: "A second methodological concern which is also related to the augmentation of the in-situ sample with literature ("secondary") data is the difference in the definition of albedo. Much of the secondary albedo data are for a broader spectral range (e.g., 295-2800 nm) than what is measured in-situ at the authors' own study sites (i.e., 300-1100 nm). This is important given the high albedo of vegetation in spectra above 1000 nm and given the sensitivity of the shortwave near-infrared broad band (1300–2500 nm) to differences in boreal tree species (see Hovi et al. 2017, https://doi.org/10.14214/sf.7753)."

Authors' Response 2: Referee #2 emphasizes that as silicon pyranometers do not

sense beyond 1100 nm, we are missing an important part of the vegetation albedo as boreal vegetation shows higher sensitivity in SWIR (1300–2500 nm) region. Boreal vegetation sensitivity in the SWIR region might be interesting for species identification based on their unique spectral signatures, but from an energy/albedo perspective vegetation albedo in this spectral region might not be as important. Firstly, there is low energy available in this region as water-vapour/aerosol related atmospheric absorption is very high in SWIR (1100–2500 nm). In winter, energy in the SWIR region is even lower, and the canopy is either leafless or a good portion it is covered with snow. Secondly, even in the growing season, depending on leaf water content, foliage absorption in SWIR and scattering in NIR regions are very high (Ceccato et al. 2000). So, it does not seem plausible that vegetation albedo in the SWIR region is a large part of the broadband albedo. Similarly, the contribution of understory vegetation in boreal forest reflectance was also found to be affecting mainly visible and NIR region (Rautiainen and Lukešd 2015). As noted in the response to Reviewer #1, simulation studies suggest a theoretical maximum deviation in albedo values between instruments based on thermopile vs. silicon pyranometers of $\sim$0.09, but this is typically < 0.05. Under field conditions even class 1 thermopile instruments show deviations of 5-7% (Stroeve et al 2005).

Referee #2 also referred to Hovi et al (2017), who report leaf-level reflectance/albedo measurements of some boreal conifer and broadleaf tree species. These measurements are important for modeling albedo using radiative transfer models in combination with other parameters such as leaf angle distribution and LAI, but leaf-level reflectance values alone do not represent stand-level vegetation albedo.

Additionally, the common species in our study plots are trembling aspen, jack pine, and black spruce. Results from Hovi et al (2017) indicate that among the species measured, leaves of trembling aspen, jack pine, and black spruce showed the least response in the SWIR region. In young (0-3 years) post-disturbance sites there were essentially no trees. So, we do not think vegetation albedo in the SWIR region is an

important source of error of silicon pyranometers in our study.

Referee Comment 3: "I also have some concern about the study's scientific value, irrespective of my concerns about the methods. None of the three major findings listed above are novel and can be distilled from a diligent review of the boreal forest albedo literature (e.g., post-fire: Lyons et al. 2008; Randerson et al. 2006; Amiro et al. 2006b; Liu et al. 2005; Wang et al. 2016 → https://www.sciencedirect.com/science/article/pii/S0034425716300888 post-harvest: Kuusinen et al. 2016; Kuusinen et al. 2014; Bright et al. 2013; Hu et al. 2018 → https://agupubs.onlinelibrary.wiley.com/doi/full/10.1029/2018MS001403 )."

Authors' Response 3: We hope that Reviewer #2 is aware that we have cited most of the articles (except Hu et al. 2018) they have listed. We have also discussed the strengths and limitations of previous studies in boreal forests and formulated research questions for our study. A diligent reviewing of the listed article may have its own scientific merit, but of course is not equivalent to collection of new field data. None of the studies listed were designed to answer the specific questions for mixedwood boreal forests we have addressed in this study, and they were concentrated either on post-fire or post-harvest sites, not on both.

Reviewer #2 suggests that there are additional data from studies in post-fire stands; however, with the exceptions of Amiro et al. (2006b) and Liu et al. (2005), the studies cited are all based on satellite data, not ground-based measurements. Similarly, all suggested studies for post-harvest stands are based on satellite data. Satellite-based albedo measurements often show biases due to atmospheric effects and angular corrections (Bright et al 2015). Due to limitations of satellite-based measurements, it is very important to have field measurements and to validate process-based hypotheses from field data.

Among the studies listed, Randerson et al. (2006), Lyons et al (2008), and Liu et al. (2005) are from the same study area, and essentially use the similar dataset to answer

different questions. Amiro et al (2006b) is the only study that presents long-term (>1 year) field data in post-fire stands. Prior field studies are also mostly limited to the summer and winter seasons, and do not present shoulder season data.

Given that we are using long-term (2013-2017) field data from both post-harvest and post-fire boreal mixedwood stands to answer specific questions, we strongly disagree with Referee #2's statement that results presented in this study are not novel. The main published study that has integrated post-disturbance data on albedo in boreal forests is that of Amiro et al. (2006), which integrates data from 22 sites and ∼37 site-years of measurements. The new data presented here are from 15 instrumented sites each monitored for 4 years, so 60 site-years of measurements, all in mixedwood boreal forests that the most important forest from a forest management perspective, but for which there are almost no prior albedo measurements.

To reiterate the main novel points of the study: i) Winter and spring albedo values are substantially higher in post-harvest than in post-fire stands.

ii) Post-disturbance patterns of recovery in albedo in boreal mixedwood stands are strongly influenced by changes in species composition.

iii) Differences in species composition were a more important driver of albedo than stand-age-related differences in boreal mixedwood stands.

iv) There are important stand-age-related dynamics in albedo in the first 15 years following disturbance events that have been "missed" by prior studies.

Referee Comment 4: "Further, the study is motivated by the need to "improve climate model parameterizations" but the authors have made no attempt to explain how their results can/will achieve this. How will the presented statistical functions or empirical insights be applied in a climate modeling context, either for improving existing parameterizations in a climate model directly or for use as a climate model benchmarking/evaluation tool? Albedo parameterizations in most climate models are processoriented and intimately tied to important forest structural attributes like leaf area in-dex which the authors have not included. Model parameterizations are also largely oriented around important local meteorological state variables (i.e., near surface air temperatures, wind speeds, precipitation type and frequency, snow depth, etc.) which are absent in the paper. This makes it difficult to discern the conditions under which the reported findings may be applied to evaluate climate model predictions. Further, since the reported albedo dynamics for the post-harvest case are intimately connected to the specific management practices of the study region, without providing any detail about the prevailing management regime(s) of the study region it will be difficult for modelers to assess accuracy of simulated post-harvest albedo dynamics. As for the post-fire case, the finding that the near-term (< 25 yr) increases in summertime albedo are connected to pioneer birch succession (a finding reported in several of the refer-ences listed above) implies that any "improvement" to the albedo prediction capability of a climate model would need to target the vegetation dynamics routines of the model and not necessarily the "albedo parameterization" itself."

Response 4: The main "motivating statement" in the paper (in the last paragraph of the introduction) reads as follows: "Deeper understanding of the local mechanisms that account for variation in albedo will not only enhance global climate models (for example, via improving the land-surface model: Bright et al., 2018), but also help to design climate-friendly silvicultural practices (Astrup et al., 2018; Bright et al., 2015a; Matthies and Valsta, 2016)." We thus think the reviewer somewhat mis-characterizes the stated motivation, which is not climate model parameterization. We will further emphasize the inclusion of vegetation dynamics in the land surface model to improve albedo prediction in our revised manuscript.

References:

Bright, R. M., Myhre, G., Astrup, R., Antón-Fernández, C. and Strømman, A. H.: Ra-diative forcing bias of simulated surface albedo modifications linked to forest cover changes at northern latitudes, Biogeosciences, 12(7), 2195–2205, doi:10.5194/bg-12-

2195-2015, 2015.

Ceccato P., Flasse S., Tarantola S., Jacquemoud S., Grégoire J. M.: Detecting vegetation leaf water content using reflectance in the optical domain. Remote Sensing of Environment 77: 22-33, 2001.

Francois, C., Ottle, C., Olioso, A., Prevot, L., Bruguier, N., and Ducros, Y.: Conversion of 400–1100 nm vegetation albedo measurements into total shortwave broadband albedo using a canopy radiative transfer model. Agronomie 22: 611–618, DOI: 10.1051/agro:2002033, 2002.

Liu, H., Randerson, J.T., Lindfors, J., Chapin, F.S.: Changes in the surface energy budget after fire in boreal ecosystems of interior Alaska: An annual perspective. Journal of Geophysical Research 110, https://doi.org/10.1029/2004JD005158, 2005.

Rautiainen, M. and Lukešd, P.: Spectral contribution of understory to forest reflectance in a boreal site: an analysis of EO-1 Hyperion data. Remote Sensing of the Environment 171: 98–104, 2015.

---

## Author Comment (AC3) · 15 Feb 2019

We truly appreciate Editor's time to read the manuscript and make critical comments.

Editor's Comment 1: "In the introduction previous reports are cited to list possible drivers. Stand age is, however, not among the listed drivers. In the discussion the authors do a reasonable good job in focusing the discussion on the physical drivers (fraction of deciduous trees, charcoal, stand structure, ...). From this perspective it is surprising that the results section uses stand age as one of the independent variables to explain the changes in albedo (as reflected in the statistical models and the table). In my opinion, the authors should better explain that the analysis with age is simply to describe the temporal evolution but that the additional analysis are intended to explain the physical drivers of these age trends. If this indeed reflects the thinking of the authors, the paper should be edited towards this message, e.g. no models should be fitted against age and several sentences throughout the manuscript should be rephrased. Nevertheless, if the authors interpret their results as an indication that age itself is a physical driver of albedo, it should be discussed how stand age (rather than structure) affects albedo."

Response to Editor's Comment 1: Thank you for your comment. In a sense this is a philosophical point: time is not itself a physical driver of any process (physical or biological), but all processes develop through time. There is a long history of quantifying patterns of stand development in forest science with time as the main independent variable: essentially all growth and yield models in applied forestry use this approach. Such models have then commonly been applied to understanding climate feedbacks.

In climate models, albedo has most often been treated as a constant for a land cover type; however, it has recently been argued that in forest systems stand age (time) should be treated as a "time dimension" to capture albedo dynamics, which also allows effective conversion to other time-dependent radiative processes (Bright et al. 2016). In part, this argument is that stand age is a more parsimonious predictor, and in part stand age may account for other co-occurring age-related factors that are not included in the model. Stand structure generally shows strongly non-linear relationships with stand age (Avery and Burkhart, 2015), and albedo models based on stand structure alone may not capture important system dynamics that are better captured using stand age. Incorporation of stand age as a predictor also allows examination of time-dependent interactions among different independent variables in empirical models.

We agree that it is important to note that stand age might not always be a good predictor of forest albedo; indeed, some prior studies have found important patterns of variation that are not well explained by stand age (e.g., Kuusinen et al. 2014; Lohila et al. 2010). These "null" results are also important contributions that require one to consider stand

age as a potential driver.

As noted above, stand age is a very commonly used metric in the forest science literature, and used as the main independent variable along with other stand structural and compositional properties to predict forest growth and yield. Researchers also have commonly used stand age to predict snow/wind vulnerability of forest stands along with other age-dependent variables (Kamo et al. 2016). Including stand age offers a simple and easy-to-measure tool to incorporate these models in forest management practices to help develop climate-sensitive forestry practices.

Editor's Comment 2: The importance of this study for climate modelling should be rewritten in line with the state of art of albedo modelling through canopy radiative transfer models and the simplified canopy radiative transfer schemes that are used in the land surface schemes of climate models. The authors seems not be aware of recent work (Naudts et al 2016, Luyssaert et al 2018) that does account for the effect of stand structure, tree species, and forest management on albedo and the climate (including not only albedo but also transpiration and roughness). The impact on modelling efforts of the albedo observations presented in this study is largely overstated. Canopy radiative transfer schemes combine scattering parameters and simulated canopy structures to simulate the albedo. The albedo values reported in this study can be used to evaluate existing models but are unlikely to be useful to improve existing models as claimed in the text. It may be best to delete all references to model developments and focus the discussion and conclusions on the underlying processes and the remaining unknowns.

Response to Editor's Comment 2: Thank you so much for suggesting two recent works. We will review and incorporate them in our revision to reflect the importance of albedo-stand age dynamics through canopy radiative transfer models.

References

Avery, T. E., and Burkhart, H. E: Forest measurements (Fifth Edition). Waveland Press, 358p, 2015.

Bright, R. M., Bogren, W., Bernier, P., Astrup R.: Carbon-equivalent metrics for albedo changes in land management contexts: relevance of the time dimension. Ecological Applications 26(6): 1868-1880. https://doi.org/10.1890/15-1597.1, 2016.

Kamo, K., Konoshima, M., Yoshimoto, A.: Statistical analysis of tree-forest damage by snow and wind: logistic regression model for tree damage and Cox regression for tree survival. FORMATH 15: 44–55, DOI:10.15684/formath.15.005, 2016. Kuusinen, N., Tomppo, E., Shuai, Y., Berninger, F.: Effects of forest age on albedo in boreal forests estimated from MODIS and Landsat albedo retrievals. Remote Sensing of the Environment 14: 145-153, 2014.

Lohila, A. K., Minkkinen, Laine, J., Savolainen, I., Tuovinen, J.‐P., Korhonen, L., Laurila, T., Tietäväinen, H., and Laaksonen, A.: Forestation of boreal peatlands: impacts of changing albedo and greenhouse gas fluxes on radiative forcing. Journal of Geophysical Research: Biogeosciences 115:1–15, 2010.

---

## Author Response (AR1)

**Point-by-point responses to the reviewers' comments**

**ANONYMOUS REFEREE #1**

**General comments**

This study investigates the effect of disturbance type (harvest, fire) on boreal forest albedo, based on in situ albedo measurements. The authors conclude that i) post- harvest albedo is higher than post-fire albedo, ii) albedo saturates at ~50 years' age after both disturbance types (which is later than the authors expected: they expected saturation at ~25 years' age or earlier), and iii) successional changes in species composition are a key driver of age-related patterns in albedo. I see high risk that the conclusions are not valid because:

**1.** The authors complement their data with data taken from previously published papers (which they call 'secondary sources'). When looking at those papers in detail it is noticed that they used pyranometers working at full range of the solar spectrum (from approx. 300 to 2800 nm), while the data collected by the authors is recorded at visible and near infrared spectral regions (300–1100 nm). This can result in substantial differences between the data sets. Another problem is that the effects that are seen may be due to differences in climate (particularly snow depth, snow properties, and extent of snow-covered period), rather than forest structure or species proportions. The most distant study site (Alaska) is thousands of kilometres away and located in different latitudes than the main study site (~65°N compared to 49.55°N). Therefore, it is likely that not only climate, but also the forest structure (e.g. height, canopy closure) as function of age, differs notably from the forests measured by the authors.

*Authors' response to #1:*

*In this revision, we have excluded data from the Alaskan site and used data only from the stands (secondary sources) that closely matched with our study sites (figures from new analyses: 1, 4, 5, 6, 7, 8). This reanalysis did not affect our conclusions drawn from the previously used dataset. Additionally, we have conducted a direct field comparison of albedo measurements from the Si-based (300–1100 nm, used in this study) and thermopile-based CNR1 pyranometers (305–2800 nm, used in the studies providing secondary data) (please see the supplementary materials). Results from this experiment indicate that the average difference in albedo estimates between these pyranometers was negligible (0.0028; not significantly different from zero). Therefore, we concluded albedo measurements from these sensors are comparable and no further corrections are necessary.*

**2.** The amount of data is relatively small (only 15 plots + those obtained from secondary sources). For example, the conclusion that wintertime post-harvest albedo is higher than wintertime post-fire albedo seems to be due to some post-harvest plots showing very high albedos. The variation in albedo among post-harvest plots is large (Figure 4a). Removing some of the post-harvest plots with high albedo would probably result in notably different conclusions. Another example is the conclusion that albedo saturates later than at 25 years' age (but at no later than 50 years' age). I think that it is impossible to make such conclusion, because there is no data measured for stands aged 20–50 years.

*Authors' response to #2:*

*Maintaining 15 plots in a spatially dispersed remote location throughout the year for 5 years is a difficult and expensive*

40  *task. This is one of the main reasons why there are not many long-term field measurements of boreal albedo.*

*Each data point in Fig. 4 is not a single measurement; it is the seasonal average albedo of a stand for the particular year – the individual sensors logged at 10-minute intervals, so each data point shown integrates thousands of individual measure-ments. In a long-term measurement, variations in stand structure and composition (particularly at the stand initiation stage)*

45  *are obvious, which in turn make albedo responses more variable. This is not uncommon in published studies (for e.g., figure 2, Amiro et al. 2006). In fact, the present study presents the single largest data set of ground-based field measurements of albedo from the mixedwood boreal region in terms of sensor-years of data.*

*We agree that there are no direct measurements between 20–50 years: this is an outstanding data gap globally. However,*

50  *looking at albedo trends with stand age from our analyses, and data from remote sensing studies suggest that albedo tends to saturate at ~ 50 years in the mixedwood boreal forest. We did not claim this is precisely 50 years: the precise wording used is "at no later than 50 years' age".*

**3.** The error sources and quality control of the measurements are not described in detail enough. The highest summer- and

55  wintertime forest albedos measured by the authors are towards the higher end of what has been reported earlier. This may be because of limited spectral range in the measurements (i.e. not a measurement error), but it is difficult to say for sure, be-cause this is not discussed by the authors.

*Authors' response to #3:*

*In this revision, we have provided the details of quality control for the measurements of irradiance and reflected solar radi-*

60  *ation in subsection 2.2 of the Materials and Methods section. This concern is also addressed in the Supplementary Materi-als of this revised manuscript.*

**4.** From 1 it follows that the secondary data sources are not comparable with the authors' measurements and should be left out. I appreciate value of in situ data and the effort that the authors have put in the experiment. It might be possible that the

65  authors' measurements alone would result in interesting conclusions. However, this is difficult to evaluate based on the data and figures presented. I provide specific comments below to help the authors improve their work.

*Authors' response to #4:*

*Thank you for appreciating our efforts to conduct this long-term field campaign. On comparability of our measurements with the secondary data, we have provided a detailed response previously during the discussion. In addition to that, we*

70  *have conducted a field comparison of the two sensor types (Si- vs. thermopile-based) used in our study and the studies providing secondary data. Results from this study indicated that measurements from these sensors are highly comparable. Please see the Supplementary Materials for full details.*

75

**Specific comments**

**5.** L17-18: It would be useful to state which stand ages these values (63%, 24%) apply.

*Authors' response to #5:*

*Thank you. For winter it's 0–19 years old stands and for spring it's 7-19 years old stands. We have clarified this in the Abstract.*

**6.** L21: It is not clear what is meant by "seasonal albedo". I suggest defining the concept before its usage.

*Authors' response to #6:*

*Here 'seasonal albedo' refers to the average albedo over a season. We substituted the wording "seasonal averages of albedo".*

**7.** L21-23: I agree that change in species composition when the forest gets older is one driving factor of albedo changes. However, also the forest structure changes when the forest gets older. For example, increasing canopy closure reduces the visibility of ground surface, and increasing tree height/canopy closure increase the shadow fraction. These both reduce forest albedo. There is lots of empirical evidence (at least based on satellite albedo measurements) in literature suggesting that albedo of coniferous forest changes with stand age, even though the species composition does not change. Thus, I think that your statement here is too strong. Species composition is one driver of age-related albedo changes, but based on the data presented, I would not say that it is a "key driver".

*Authors' response to #7:*

*We agree that species composition is 'a key driver' not 'the key driver'; we have "tweaked" the wording in the manuscript at a couple points to be clear on this point. Our data suggest that species composition is a better predictor than stand age and other stand structural properties: overall lower RSE of regression models with % deciduous broadleaf species as a predictor (Fig. 6) compared to models with other stand characteristics (Fig. 5, 7, 8). Here we would also like to emphasize that this study is in a mixedwood boreal forest (in mesic sites) which has different successional pathways than a pure jack pine (in dry sites) or black spruce (in wet sites) stand. In these pure boreal stands, it is very common that species composition does not change with stand age, but for a mixedwood stand that is uncommon.*

**8**. L34: It might be useful to clarify what is meant by "the relative stability of the atmospheric temperature profile".

*Authors' response to #8:*

*We have revised the wording as follows to be more precise: "…the relative stability of the atmospheric temperature profile due to weak latent-heat-driven convection".*

**9.** L70-72: I doubt that the legacy charcoal from fires that happened several years or decades ago would influence albedo at the time of harvest.

*Authors' response to #9:*

*Charcoals (and other forms of black carbon) are highly recalcitrant and can persist for a very long time (~1000+ years) in boreal soils. Because of soil scarification during harvesting, legacy charcoals from the previous wildfire can easily be exposed and can reduce surface albedo. Over the last century, fire frequency has increased so much that average fire cycle in*

*the Canadian boreal forest is now ~80 years compared to the previous ~150 years. In this scenario, it is not surprising that a post-harvest site had a wildfire some 100 years ago. We did notice frequent legacy charcoals in essentially all recent post-harvest stands. Another PhD project is currently analyzing samples from the project sties to quantify legacy charcoal abundance.*

**10.** L73-75: This sentence is also a bit unclear. It needs more clarification how (through which physical mechanisms?) decomposition processes and plant colonization would influence albedo.

*Authors' response to #10:*

*We have clarified the sentence.*

**11.** L96-97: The first hypothesis is a bit contradictory to what is stated on L65-68 (that post-harvest stands typically have higher proportion of broadleaved species). Large proportion of broadleaved species should lead to higher albedo.

*Authors' response to #11:*

*Post-fire stands can have higher proportions of deciduous broadleaf species depending on legacy stand characteristics and fire frequency/intensity. In our study sites, however, we had a higher percentage of deciduous broadleaf species in the post-harvest stands. We have revised the sentence to avoid the confusion.*

**12.** L105: I think it is important to explain (in qualitative, descriptive terms) what kind of structure do the post-fire stands have. Do they have lots of standing dead trees? How severe were the fires?

*Authors' response to #12:*

*Fires in all sites (including secondary data sites) were severe which killed all previous vegetations. We have explained this in more detail in subsection 2.3 of Materials and Methods.*

**13.** L107: It is not clear whether all three replicate plots were in separate stands or in the same stand?

*Authors' response to #13:*

*The replicated plots were in separate stands. We have revised the text to make this point more explicit.*

**14.** L111-119: Would it be possible to show the forest structural variables for each study plot in a table? This way the reader would get better understanding of the forest structure and species composition and their development through stand age.

*Authors' response to #14:*

*We already had some information on forest structural variables in subsection 2.1 of the Materials and Methods section. We have added Table 1 to provide more general information on these variables.*

**15.** L117: What does the abbreviation LFH stand for?

*Authors' response to #15:*

*'LFH' stands for 'Litter, Fermentation, Humus'–a common acronym used in soil science literature to describe soil organic horizons develop in forests. We have elaborated the abbreviation in parenthesis.*

**16.** L121-124: How was the placement of the albedo measurement towers? I guess that the stands were surrounded by older (higher) forests? How far from the stand edges the towers were placed? Did the surrounding forests block a portion of the incoming diffuse radiation and is there a possibility that this would have affected the measurements?

*Authors' response to #16:*

*The selected stands were at least 5 ha in size and the towers were set up in the middle of the stands. There were no old-er/taller forests within a few hundred meters in every direction. So, there was no possibility of any older/taller forest being within the footprint of the pyranometers or blocking incoming solar radiation. We have added more explanations in the subsection 2.2 of the Materials and Methods section, where we write: "Selected stands were at least 5 ha in size, and plots were established at least 100 m from any older or taller stand to avoid edge effects".*

**17.** L122: Due to limited spectral range (300–1100 nm) the upper end of solar spectrum (from 1100 nm up to 4000 nm) is left out and therefore the measured albedo is not full shortwave albedo. I looked at the methods of the papers providing secondary data sources, and noticed that they used full solar spectrum: -Chambers and Chapin (2002), Liu et al. (2005): Eppley precision spectral pyranometer, 285–2800 nm -Amiro et al. (2006a): Kipp and Zonen CNR1, 305 to 2800 nm -Amiro et al. (2006b): Kipp and Zonen CM3, 305 to 2800 nm

*Authors' response to #17:*

*We have responded to this query previously as a 'short response' during the discussion (also, please see the Supplementary Materials)*

**18.** L126-127: What kind of quality control procedures were applied in the data processing? The explanation in the manuscript gives an impression that data from all days were useful and no outliers etc. needed to be removed. Did you remove some observations/days due to low quality?

*Authors' response to #18:*

*We followed the standard protocols for data quality control, and have added a description of the quality control procedure in the 'Experimental Setup" subsection of the "Materials and Methods".  The full paragraph read as follows:*

*"Quality control for the irradiance and reflected solar radiation measurements was conducted following guidelines of the World Meteorological Organization (WMO). Any unusually high/low values were replaced by interpolated values by taking the average of preceding and subsequent measurements. Daily total irradiance data were compared against the WMO-provided maximum possible daily sums of clear-sky irradiance for 50°N latitudes (Annex V, Page 26, WMO 1986). If measured the daily total irradiance was higher than the maximum possible value, we excluded the measurements for that day. For reflected solar radiation, if the daily total of reflected solar radiation was higher than the daily total irradiance, we also excluded the measurements for that day.  In addition, we excluded measurements for any snowy day; snowfall was detected using data from the closest available weather station (Environment Canada, 2018)"*

**19.** L131: "deciduous broadleaved area". Does this refer to basal area, or something else?

*Authors' response to #19:*

*Yes, it refers to the percentage of deciduous-broadleaf basal area of the total basal area. We had already explicitly mentioned this in the Materials and Methods section (which states: "The proportion of deciduous broadleaf species of a plot*

*was calculated as the ratio of basal area of the deciduous species to the total basal of area of the plot"), so no change seems necessary.*

**20.** L132-133: The diameter limit is much stricter than the height limit. Usually trees with 5 cm diameter at breast height have height at least ~5 meters. This means that you do not need the height limit at all, because the diameter limit already excludes all trees with height less than 5 m. The diameter limit is also quite high considering the young age of the forests. I do not know exactly how fast the trees grow in the study area, but I would guess that forests with ages from 0 to 10 years have only few trees (if any) with diameter at breast height exceeding 5 cm. Is the high diameter limit the reason why some stands are missing in Figure 6 (that shows the albedo dependence on broadleaved proportion)?

*Authors' response to #20:*

*Trees in the study area actually grow relatively rapidly, but generally have a low height-diameter ratio (5-cm DBH trees are only about 2-3 m in height). In the study area, stand density for the 0–19 years old stands averaged ~7000 stem/ha. We have included information on stem density in Table 1.*

*Post-harvest stands with high albedo were 0–4 years old, which had only a few seedlings of deciduous broadleaf species. If we included those seedlings, the % deciduous species became 100%, which was misleading (compared to other plots, they were not zero either). These youngest stands were excluded from the % deciduous broadleaf species calculation and were included as ground vegetation cover. Additionally, % deciduous broadleaf species of some secondary-data sites were not reported, so were excluded from the analysis. That is why some stands are missing in Figure 6.*

**21.** L137-140: Which of the measurement years (2013-2017) were used in calculation of the age? Or did you treat each year as separate observation and thus the age differed depending on which year was used?

*Authors' response to #21:*

*Stand age was calculated as the difference between the year of disturbance (fire/harvesting) and the year of measurement (2013–2017). So, yes, we treated each year as separate observations and stand age was based on the year of measurement.*

**22.** L143-153. How does the climate in the secondary sites (particularly snow depth, snow properties, extent of snow-covered period) differ from the site in Ontario? This is very important because if the climate differs markedly, then the observed differences between post-fire and post-harvest stands are not solely due to stand structure and species composition.

*Authors' response to #22:*

*Weather data from Environment Canada (2018) indicated that climate conditions of the secondary data sites are similar to our study sites. In all these sites average snow duration is 5–6 months. We do not have snow property data, but average snow depth in the Lake Nipigon study area is ~ 11cm, and in Saskatchewan and Manitoban sites 10–15 cm. So, we do not expect large differences in albedo due to geographic differences in the snow regime.*

**23.** L165: "top-surface-specular-included (diffuse and direct) reflectance" is a bit awkward definition. If the reflectance values were measured with integrating sphere setup (collimated light used for illumination, and the reflected radiation collected over the hemisphere), then I would express it something like: "directional-hemispherical reflectance factor of the top-surface of the soil sample".

*Authors' response to #23:*

235 *Thank you. We have revised the text according to your suggestion.*

**24.** L169: Does "ten different locations" refer to ten different locations within the sample? How is a location defined (how large area is covered by one measurement)?

*Authors' response to #24:*

240 *Yes, ten different locations of a single sample to accommodate spatial heterogeneity in reflectance of the sample. The size of each location was dependent on the diameter of the opening of the integrating sphere on the spectrometer, which was 10.32 mm. We have added this detail in the methods as follows: "Every sample was measured ten times in ten different locations (each 0.84 cm$^2$ in area) …".*

245 **25.** L169: What does "Boxcar width 5" mean? It needs more explanation.

*Authors' response #25:*

*Boxcar is a spatial averaging method to remove noise by averaging reflectance/absorbance values of adjacent pixels (of the Charge-Coupled Device [CCD]) and hence improves the signal-to-noise ratio. Here, by "Boxcar width 5" we mean that reflectance values were calculated by spatial averaging of 5 CCD pixels (2 to the left + 1 center + 2 to the right) to enhance*
250 *signal-to-noise ratio. We have added an explanation in the manuscript as follows: "(with Boxcar width 5 [spatial averaging of 5 pixels] and 100 millisecond integration time)".*

**26.** L187-196: Are the selected model shapes based on physical nature of the phenomena studied, or are they just chosen to give the best model fit to the observed data? For example, in Figure 6g it is difficult to imagine a physical reason why albe-
255 do would first increase as function of broadleaved percentage, and then decrease again as the broadleaved percentage approaches 100%.

*Authors' response to #26:*

*We used a purely statistical basis for model selection (i.e., minimum AIC), but we tested candidate models that potentially provide a good description of physical processes. The patterns presented in Fig. 6 show a saturating response in all but one*
260 *case, and was our expectation based on physical processes. We agree that the decline in albedo shown in figure 6g (fall patterns for post-harvest stands) at high proportions of deciduous species is not intuitive. However, this could be related to leaf senescence patterns in conjunction with leaf litter reflectance properties: in particular aspen leaf litter is quite black, and if not covered by snow could certainly reduce albedo in nearly pure aspen stands in the fall.*

265 *To address this point, we have added the following to the results section: "In the case of fall albedo in post-harvest stands, there is an apparent decline in nearly pure stands (Fig. 6g), with a better fit of the double exponential model. We speculate that very dark post-senescence leaf litter of aspen may account for this effect".*

**27.** L209-210: Variability of albedo in post-harvest stands in winter and springtime is indeed very high. For example, in
270 Figure 4a it is seen that the albedos of the young forests (<50 years) vary from approx. 0.2 to almost 0.9. I think that 0.9 is very rarely observed except for pure snow surfaces with no vegetation. Are you sure that the variation is not caused by measurement errors? This is why I suggest reporting the details of your measurement and quality control procedure in detail. Another thing that caught my attention is the high albedo (approx. 0.3) of some stands in summertime (Figure 4c, d).

Albedo values 0.2 (at solar noon) are rarely reported in boreal forests. Your values are approx. 50% higher than that. Is this because of the limited spectral range of the pyranometer?

*Authors' response to #27:*

*Some of the youngest stands had very little vegetation, and had prolonged periods of 100% show cover. In this revision, we have provided detailed information on measurement quality control in the Materials and Methods section. We have also provided data from field measurements comparing performances of both Si-based and high-end pyranometers (please see Supplementary Materials). On albedo variances, we have discussed in Authors' response to #2.*

**28.** Figures 2, 4–8: The number of observations varies between figures. It is not clear what constitutes an observation? There were 15 plots, but the number of observations can be much higher than this. Is each year treated as a separate observation? Text on line 128 suggests this. If each year is treated as a separate observation, how does it affect/violate assumptions (independence of observations) in the statistical models?

*Authors' response to #28:*

*This is a good point. We treated seasonally averaged measurements from the same stands as independent measurements, so the reviewer is correct that this does raise issue of non-independence of measurements – as might be handled by treating observation sites as a random effect in a linear mixed effects model. For the general linear models (i.e., ANCOVAs as described in methods) we have conducted parallel analyses that incorporate a random effect for stand. These analyses yield essentially identical results to those presented, and in all cases the random effect term is not significant. We note these parallel analyses in the methods section but present only the (simpler) analyses.*

**29.** Section 3.2: It might be possible to weight the observed ground spectra with incoming solar radiation, to calculate albedos of the ground surface. This way, it would be easier to link age-related changes in ground albedo to age-related changes in forest albedo.

*Authors' response to #29:*

*Albedo of the ground surface depends on many factors (moisture content, presence/absence of trees etc.) in addition to incoming solar radiation. Stand structure and species composition are also likely to have strong interactive effects on ground surface albedo. We respectfully argue that this would not be a viable approach to use any simple weighting scheme to separate ground albedo from vegetation albedo effects.*

**30.** Figure 6: Why Figure 6 does not contain all data presented in Figure 4 (high winter albedo values close to 0.9 are not presented in Figure 6)?

*Authors' response to #30:*

*The youngest stands (without enough trees to estimate proportion of deciduous species) were omitted from these analyses. Please see response to # 20*

**31.** L254-255: Earlier (in Section 2.5) you state that only fall albedos were modelled with double-exponential model, but here you say that also summer albedo was modelled with double-exponential model. Which one is true?

*Authors' response to #31:*

*In section 2.5, at L192–194 we were describing why we could not use generalized linear models for some stand attributes to predict fall albedo, as they were only related to fall albedo nonlinearly (double exponential). If we included these stand attributes with other attributes in a generalized linear model (GLM), the model structure became very complex (a mixture of*

315 *linear and non-linear families). An immediate challenge of this modeling exercise was to decide how to define a GLM family that can handle both double exponential and exponential families at the same time. To avoid this complex modeling exercise, we chose to model these stand attributes separately. In L254–255 we were describing how summer albedo was related to the stand age only (not in the GLM with other stand attributes).*

320 *So, to summarize, both statements are true and are referring to two different contexts as discussed above.*

**32.** L275: Please explain in more detail what is meant by "to avoid modelling complexities". Does it mean that the model did not converge if it was too complex?

*Authors' response to #32:*

325 *Please Authors' response to #31*

**33.** L287-288: I think word "dramatic" is too strong.

*Authors' response to #33:*

*A difference in albedo of >0.2 is really large: larger than differences between, say, generalized albedo values typically giv-*

330 *en for grasslands (~0.25) vs. forests (0.08-0.18). It seems essential to emphasize the surprising magnitude of the differences an albedo values with disturbance type as described in the study.*

**34.** L290: I doubt that the conclusion that albedo saturates at 50 (rather than at ~25) years' age is valid. You do not have any measurement data between 20-50 years (Figure 4).

335 *Authors' response to #34:*

*Please see Author's response to #2.*

**35.** L291: Please explain in more detail how you determine that the effect of broadleaved proportion is larger than the stand age effect?

340 *Authors' response to #35:*

*This conclusion is based on the lower RSE values of the regression models for broadleaf proportion compared to other regression models. Also, please see Authors' Response #7.*

**36.** L306-308: I think this sentence is a bit misleading. I would expect that from an energy balance perspective late spring is

345 more important than winter, because the amount of incoming solar radiation is larger in late spring.

*Authors' response to #36:*

*Snow starts melting in late spring because of high incoming solar radiation. Melting snow is usually dirty and has lower albedo compared to regular snow (as reported, for example, in Conway et al. 1996). During this time of the year, latent heat flux from the melting snow usually very strong and thus dominates the surface energy balance.*

350

**Technical corrections**

**37.** L177-178: Near-infrared range must have a typo [700–100 nm].

*Authors' response to #37:*

*Yes, it should be [700-1000 nm]. Thank you. We have corrected this typo.*

**38.** L206-207: "post-harvest" repeated two times.

*Authors' response to #38:*

*Thank you. Corrected.*

**39.** Table1: It is a bit difficult to see the difference between italic vs. regular font. Perhaps bold vs. regular would be a better choice?

*Authors' response to #39:*

*Thank you. We changed italic fonts to bold.*

**40.** L357: "albedo" repeated two times.

*Authors response to #40:*

*Thank you. Corrected.*

**ANONYMOUS REFEREE #2**

By combining literature data with novel in-situ measurements and via chronosequencing, the study by Halim et al. analyzes temporal trajectories in surface albedo following harvest and fire disturbances in southern boreal mixedwood forests. The main findings are that i): winter and spring surface albedos following harvest disturbances are higher than those following fire disturbance; that ii) both winter and summer surface albedos "saturate" at around 50 years, and that iii) successional changes in species composition are a key driver of post-disturbance albedo dynamics.

**1.** I have several major concerns about the study methods that call to question these findings. My first and largest concern surrounds the extensive use of albedo data sourced from the literature (referred to as "secondary data") which are connected to sites located hundreds to thousands of kilometers away. Although the dominant species com- positions across sites may be similar, stand structure and other important site-specific attributes affecting the surface albedo may differ greatly across sites. These include differences in geology and soils (affecting albedo via their controls over understory vegetation composi- tions, soil moisture retention, growth rates), differences in latitude (affecting the direct albedo component via differences in solar geometry), and – most importantly – differences in local climate (affecting albedo via controls over soil moisture, veg- etation growth and phenology, length of snow season, and important snow physical attributes such as snow depths, snow age, snow water contents). Without controlling for differences in these important site-specific factors it is difficult to arrive at robust conclusions regarding albedo-age dynamics, albedo-species composition dynamics, albedo-canopy height dynam- ics, and albedo-ground cover dynamics. Regarding the albedo-age dynamic, for instance, asymptotes of the presented expo- nential models in Figure 4 seem to be heavily influenced by the "secondary" data comprising all data points beyond 19 years. Regarding the albedo-species composition dynamic, the "secondary" data points in Figure 6 for "Summer" and "Win- ter" seem to be heavily influencing the y-intercepts and thus affecting the model functional form and shape parameters. Secondary data points in Figure 7 also appear to heavily influence the model fits (or lack thereof) for the "Summer" albedo-canopy height dynamics.

A second methodological concern which is also related to the augmentation of the in-situ sample with literature ("second- ary") data is the difference in the definition of albedo. Much of the secondary albedo data are for a broader spectral range (e.g., 295-2800 nm) than what is measured in-situ at the authors' own study sites (i.e., 300-1100 nm). This is important giv- en the high albedo of vegetation in spectra above 1000 nm and given the sensitivity of the shortwave near-infrared broad band (1300-2500 nm) to differences in boreal tree species (see Hovi et al. 2017 → https://doi.org/10.14214/sf.7753 ).

I also have some concern about the study's scientific value, irrespective of my concerns about the methods. None of the three major findings listed above are novel and can be distilled from a diligent review of the boreal forest albedo literature (e.g., post-fire: Lyons et al. 2008; Randerson et al. 2006; Amiro et al. 2006b; Liu et al. 2005; Wang et al. 2016 → https://www.sciencedirect.com/science/article/pii/S0034425716300888 post-harvest: Kuusinen et al. 2016; Kuusinen et al. 2014; Bright et al. 2013; Hu et al. 2018 → https://agupubs.onlinelibrary.wiley.com/doi/full/10.1029/2018MS001403 ).

*Authors' response to #1:*

*We responded to these concerns previously as a 'short response' during the discussion/review process. In response to Re- viewer #1's comments, we have also discussed in detail above. For measurement comparability between Si-based and ther- mopile-based pyranometers, please see the Supplementary Materials.*

**2.** Further, the study is motivated by the need to "improve climate model parameterizations" but the authors have made no attempt to explain how their results can/will achieve this. How will the presented statistical functions or empirical insights be applied in a climate modeling context, either for improving existing parameterizations in a climate model directly or for use as a climate model benchmarking/evaluation tool? Albedo parameterizations in most climate models are process-oriented and intimately tied to important forest structural attributes like leaf area index which the authors have not included. Model parameterizations are also largely oriented around important local meteorological state variables (i.e., near surface air temperatures, wind speeds, precipitation type and frequency, snow depth, etc.) which are absent in the paper. This makes it difficult to discern the conditions under which the reported findings may be applied to evaluate climate model predictions. Further, since the reported albedo dynamics for the post-harvest case are intimately connected to the specific management practices of the study region, without providing any detail about the prevailing management regime(s) of the study region it will be difficult for modelers to assess accuracy of simulated post-harvest albedo dynamics. As for the post-fire case, the finding that the near-term (< 25 yr) increases in summertime albedo are connected to pioneer birch succession (a finding reported in several of the references listed above) implies that any "improvement" to the albedo prediction capability of a climate model would need to target the vegetation dynamics routines of the model and not necessarily the "albedo parameterization" itself.

*Authors' response to #2:*

*We also responded partly to this concern during the discussion. In this revision, we have revised the Discussion (subsection 4.2) to focus on "improving albedo prediction capability' than on "albedo parameterization". We have also provided information on current forest management practices in the study area (Subsection 2.1), as suggested by the reviewer.*

**3.** Given my concerns about the study's methods and low scientific significance, I find it difficult to recommend publication in BG. I also find it difficult to encourage a major revision involving a new analysis that excludes the use of "secondary" data given the limited number of field plots and given the narrow spectral band of albedo data that has been measured at those plots.

*Authors' response to #3:*

*Please see points made above: the present study more than doubles that number of sensor-year measurements available for field measurements of albedo in boreal forests, and we show that there is no detectable difference in albedo measurements made by the pyranometer used and the measurements based on broadband pyranometers (that have high energy demands and would be exceedingly difficult and expensive to deploy and maintain in remote field sites).*

**EDITOR'S COMMENTS**

**1.** Halim and co-authors present an interesting analysis on the age and species effect on albedo.

In the introduction previous reports are cited to list possible drivers. Stand age is, however, not among the listed drivers. In the discussion the authors do a reasonable good job in focusing the discussion on the physical drivers (fraction of deciduous trees, charcoal, stand structure, ...). From this perspective it is surprising that the results section uses stand age as one of the independent variables to explain the changes in albedo (as reflected in the statistical models and the table). In my opinion, the authors should better explain that the analysis with age is simply to describe the temporal evolution but that the additional analysis are intended to explain the physical drivers of these age trends. If this indeed reflects the thinking of the authors, the paper should be edited towards this message, e.g. no models should be fitted against age and several sentences throughout the manuscript should be rephrased. Nevertheless, if the authors interpret their results as an indication that age itself is a physical driver of albedo, it should be discussed how stand age (rather than structure) affects albedo.

*Authors' response #1:*

*In our previous 'short response' during the discussion, we discussed why stand age is an important predictor in addition to stand structure and composition (regardless of their dependency on stand age).*

**2.** The importance of this study for climate modelling should be rewritten in line with the state of art of albedo modelling through canopy radiative transfer models and the simplified canopy radiative transfer schemes that are used in the land surface schemes of climate models. The authors seems not be aware of recent work (Naudts et al 2016, Luyssaert et al 2018) that does account for the effect of stand structure, tree species, and forest management on albedo and the climate (including not only albedo but also transpiration and roughness). The impact on modelling efforts of the albedo observations presented in this study is largely overstated. Canopy radiative transfer schemes combine scattering parameters and simulated canopy structures to simulate the albedo. The albedo values reported in this study can be used to evaluate existing models but are unlikely to be useful to improve existing models as claimed in the text. It may be best to delete all references to model developments and focus the discussion and conclusions on the underlying processes and the remaining unknowns.

*Authors' response #2:*

*We agree that the relevance of direct surface albedo measurements is mainly to land surface schemes and sub-models in the climate modeling community. Existing efforts, including the recent papers cited, are limited by data availability, given the lack of data from mixedwood boreal forests and for harvest impacts as distinct from fire. The link to modeling efforts is made mainly in the final paragraph of the paper. We have revised the final paragraph so as to emphasize model evaluation. However, one would hope that there is ultimately a constructive feedback between modeling and empirical studies.*

*The revised final paragraph reads as:*

[revised manuscript text omitted]

---

## Author Response (AR2)

**POINT BY POINT RESPONSE TO THE REVIEWERS' COMMENTS**

**REFEREE #1**

**Comment 9.** The reply does not address the point made by the referee: is there robust evidence that charcoal affects the background albedo during a harvest decades after a fire? The referee seems to question this hypothesis because within days to weeks after a harvest the residual charcoal may be covered by snow, litter, herbs, mosses and/or ferns.

*Response to comment 9:*

Ground vegetation in recently disturbed sites in the study area is characteristically sparse (see photo below of a stand one-year post harvest). In older stands (45+ years post-disturbance) typical ground vegetation cover values, including non-vascular plants, are ~45% (Kumar et al., 2018).

Distinguishing charcoal from other dark material (e.g., leaf litter, organic matter, and fungi) is extremely difficult at a large spatial scale; we aware of no existing practical methods that would permit such measurements. We did not quantify the percentage of surface charcoal present at the spatial scale measured by sensors in this study (~10-100 m^2, depending on tower height). However, we did report our general field observations during data collection, in which we observed legacy charcoal as a ubiquitous component of soil surface material in both recent post-harvest and post-fire stands that could potentially affect surface albedo. Prior studies have also found that legacy charcoals can be visually observed on the forest floor and within the top 0.5 m soil many years (decades to 100+ years) following wildfire in boreal forests in Scandinavia (Ohlson et al., 2009) and Russia (Wallenius, 2002). Simulations from Tinker and Knight (2001) suggest that residuals from a fire event returning at a 100-year interval can cover ~25% of the forest floor.

In response to this criticism, we visually assessed our retained surface samples of intact forest floor (used for spectroscopic measurements: i.e., Fig. 3) for visually detectable charcoal fragments. Among post-fire stands, 100% (n = 30; surface area of each sample: 78.5 cm$^2$) samples had visually detectable charcoal; among post-harvest stands, 70% (19 of n = 27 samples – with a similar

percentage in 4-year-old post-harvest stands, and 11-19-year-old stands) had visually detectable charcoal. These data are briefly reported in results in the revised manuscript, and confirm our general field observations that charcoal is a ubiquitous component of the soil surface in the study system.

[Figure]

*Stand at 1-year post harvest (Photography: Md Abdul Halim, 4 June 2014)*

[Figure]

*Examples of surface soil samples from harvested stands with visible charcoal fragments outlined.*

**Comment 19.** This confused the referee and is likely to confuse other readers as well. Rewrite this sentence (it requires adding a single word which would have been less work than the reply and my subsequent comment).

*Response to comment 19:*

DONE: thank you. We have added the word "basal" in the sentence which now reads "… proportion of deciduous-broadleaf basal area (%), …".

**Comment 20.** Add this information to the caption of the relevant figures. Readers should easily understand why different figures have different numbers of observations.

*Response to comment 20:*

DONE: thank you. We have added a clarification note on why Figure 6 has different number of observations in the figure caption.

**Comment 26.** Replace "… aspen may account for this effect" by "aspen may cause for this effect".
*Response to comment 26:*
DONE: thank you.

**Comment 27.** Neither the reply nor the manuscript addresses why the albedo values seem 50% higher as noted by the referee.

"Another thing that caught my attention is the high albedo (approx. 0.3) of some stands in summertime (Figure 4c, d). Albedo values 0.2 (at solar noon) are rarely reported in boreal forests. Your values are approx. 50% higher than that. Is this because of the limited spectral range of the pyranometer?"

*Response to comment 27:*

Sorry, we missed this part of the comment in our prior response. As addressed in the first response, the cross-calibration results reported in the prior response indicate there was no detectable bias in albedo values estimated by the Si-based instruments used compared to a thermopile instrument.

The high summer albedo values observed correspond to a subset of post-fire stands with a high proportion (>60%) of deciduous species (mostly aspen): see Fig. 6f. There are only a few prior ground-based estimates of boreal forest albedo for deciduous-dominated stands, and one can expect variability from stand to stand. Although we did not quantify this effect, aspen in some of our sites had high levels of aspen serpentine leaf miner. Leaves affected by this insect turn nearly white in color (see photo below), and we strongly suspect this is responsible for the high summer albedo values observed in two of the stands.

[Figure]

*Aspen leaves impacted by aspen serpentine leaf miner (Phyllocnistis populiella)*

We have added a note on this pattern in the Results section.

**Comment 30.** Add this information to the caption of the relevant figures. Readers should easily understand why different figures have different numbers of observations.

*Response to comment 30:*
DONE: thank you. We have added a clarification note in figure caption on why the figure has different number of observations compared to other figures.

**Comment 31, 32, 35, 36.** This confused the referee and is likely to confuse other readers as well. Address this issue in the manuscript.

*Response to comment 31–32:*

DONE: we have rewritten the text in sections 3.3.3 and 2.5 (now 2.6) to enhance clarity.

*Response to comment 35:*

DONE: we have clarified "how we determined the effect" in the Discussion section (L335–337).

*Response to comment 36:*

DONE: we have edited the text to avoid confusion on energy balance in late spring (L350–355).

**Comment 33.** "Dramatic" is a subjective interpretation. Remove from the manuscript or define when and why a chance in albedo should be considered dramatic.

*Response to comment 33:*

DONE: we have replaced the word "dramatic" with "significant".

**REFEREE # 2**

**Comment 1.** Based on the presented material the referee concluded that the manuscript lacks novel ideas or data. The authors could add a paragraph in which they show how their work goes beyond the work presented in the papers listed by the referee. Such a paragraph should avoid that other readers come to the same conclusion as the referee.

*Response to Comment 1:*

We have responded on the data quality previously: in the revised manuscript we added results from a field calibration study that showed there was no statistically significant difference in albedo measured from Si-based pyranometers compared to that from thermopile pyranometers.

In this revision, we have included a paragraph (first paragraph of the Conclusions) that elaborates on the novel contributions of this study. In addition, we have added/modified a few sentences in the Introduction to clarify the novel points addressed.

**EDITOR**

**Comment 1.** The discussion is still very much centered on stand age and does not even touch on the issue that age is not a driving process. The age-oriented approach is a shortcoming of the study and limits the value of the results and its applicability for data-model comparison. It is not because there is long tradition of studying age-related patterns in forestry that this tradition is well justified and should not be questioned.

*Response to comment 1:*

We strongly disagree that the focus on age-dependent patterns in albedo is a "shortcoming" of the study: it is rather the main point of the study. We emphasize this in the paper not because there is a long tradition of studying age-related patterns (in forestry and other disciplines), but because assessing the larger-scale implications of ground-based measurements demand this approach. We do not see how it is possible to provide any integrated assessment of forest management impacts on albedo (or other critical climate feedbacks, such as forest carbon sequestration), without quantitatively assessing patterns through stand age. It is reasonable to describe albedo as (for example) a function of stand structure and composition (we do this explicitly in the paper). But such descriptions would only be sufficient (for purposes of modeling or integrated assessment over larger spatial or temporal scales) if one could strictly assume that the measured parameters describing stand structure and composition completely account for variation in albedo. We do not think such an assumption is warranted. Moreover, even if one did have a comprehensive physical process model that would allow prediction of albedo from, say, forest structure, understanding and modeling the implications of this from a forest management perspective (or in the context of a natural disturbance regime) would demand analyses based on temporal dynamics following disturbance.

We include the following sentences in the first part of the last paragraph of the Introduction to more explicitly justify our focus on temporal dynamics:

*"The temporal dynamics of stand albedo following disturbance regimes have critical implications to interactions between climate forcing, forest management, and disturbance regimes. For*

*example, if harvested stands converge in albedo with older stands within a few years (a small fraction of total rotation length), forest management is expected to have little impact on albedo at the landscape scale. Conversely, slow recovery in albedo or persistent effects of harvest compared to natural disturbance would indicate that the forest management regime fundamentally alters albedo-related climate feedbacks."*

We have also added a sentence in the Discussion (sub-section 4.2) that touches upon this issue.

[revised manuscript text omitted]

---

## Author Response (AR3)

**POINT BY POINT RESPONSE TO THE REVIEWERS COMMENTS**

**Response to Reviewer's comment:**

We thank the Reviewer for their specific suggestions to improve the manuscript. In this revision, we have re-analyzed the dataset after excluding the data from secondary sources and presented the results as Supplementary Table 1. We have also updated the relevant Methods and Results sections. As suggested, we have reproduced Figures 6–8 with color-scaling for stand age, demonstrating the effects of stand age on albedo, and revised the Discussion section accordingly.

**Response to Editor's comment:**

We appreciate the Editor's thoughtful suggestions to improve the manuscript. In addition to the revisions mentioned above, we have conducted analyses to show the relationships between stand age and stand structural attributes used in this study and presented the results as Supplementary Figures 2–5. We have also revised the Discussion to reflect our new analyses.

[revised manuscript text omitted]

---

## Author Response (AR4)

**POINT BY POINT RESPONSE TO THE REVIEWERS COMMENTS**

**Response to Editor's comment:**

Thank you. We have edited the Photo S1 and its caption (in Supplementary Materials) to reflect your suggestions. We truly appriciate your (and other reviewers') insightful comments, which immensely helped to improve the manuscript.

[revised manuscript text omitted]

**M. A. Halim et al.**

**Correspondence to:** M. A. Halim (abdul.halim@mail.utoronto.ca)

**Field comparisons of silicon-based pyranometers and thermopile pyranometers for**
**land surface albedo measurements**

**S1. Background**

In *"Stand age and species composition effects on surface albedo in a mixedwood boreal forest"* we use silicon (Si) photo-cell-based pyranometers (Hobo: Onset Computer, Massachusetts, USA) (spectral range: 300–1100 nm; measurement range: 0–1280 $Wm^{-2}$) to measure albedo of mixedwood boreal stands in post-fire and post-harvest chronosequences. Most prior published albedo measurements used thermopile pyranometers with a broader spectral range (~300–2800 nm). Although the narrower spectral range of Si-based pyranometers might result in lower estimates of total energy flux, potential biases in albedo estimates are less clear, and direct performance comparisons of both sensor types are very few (Dirmhirn, 1968; François et al., 2002; Stroeve et al., 2005). Direct field comparisons of the Si-based pyranometers with thermopile pyranometers are not available. In the study, we also used published albedo values (secondary data) from recent post-fire sites of similar stand structure and composition and climate, and data from the old (> 70 years) boreal jack pine (*Pinus banksiana*) stands to model trends in albedo change with changing stand age, structure, and composition. Studies providing secondary albedo data used thermopile-based pyranometers (Kipp and Zonen CNR1 and Eppley precision spectral pyranometer).

Here we present results of a supplementary calibration study conducted over nine days under variable sky conditions (% cloudiness) and ground cover (snow cover) conditions, to assess the relative performance of Si-based Hobo pyranometers in comparison to thermopile pyranometers.

**S2. Materials and Methods**

We deployed two pairs (one pair upfacing and one pair downfacing) of Si-based Hobo pyranometers at a similar height (~2 m) to a CNR1 net radiometer (Kipp and Zonen, The Netherlands) on 21st February to 3rd March 2019 at the Elora Research Station, Guelph, Ontario (43.64º N, 80.41º W) (Photo S1). Si-based pyranometers were set to measure solar radiation at 10-min intervals (same intervals used in the main study) and the CNR1 logged measurements at 30-min intervals. Out of the 11-day measurements, we excluded measurements of two snowy days—the same filtering scheme used in the main study. Over the selected nine days, sky cloudiness varied from 20–100% and albedo varied from 0.29–0.88 (because of varying snow cover conditions). Weather data was collected from the closest (within a km) Environment Canada weather station (Environment Canada, 2019). This Elora site is a post-harvest cornfield where some corn stalks are protruding through the snow cover, closely analogous to our recent post-harvest sites. We specifically chose an open site to test the performance of Si pyranometers in high snow-covered ground conditions, since studies have reported the greatest divergence in measurements between Si-based and thermopile pyranometers under conditions of high snow reflectivity (Dirmhirn, 1968; Stroeve et al., 2005).

One of the two pairs of (up/down-facing) Si pyranometers were old (used in the field for about a year) and the other pair was new (never used in the field); this enabled an evaluation for possible performance degradation due to field usage. Since CNR1 is a net radiometer, for this comparison, we only used data from the up- and down-facing CM3 modules (spectral range 305–2800 nm, measurement range: 0–1000 Wm$^{-2}$). The CNR1 net radiometer unit used in this study was factory calibrated approximately two months prior to the measurements.

Incoming/reflected solar radiation measured by both (Si-based and CNR1) pyranometers were averaged over one hour for hourly comparisons. Mean values for hourly average incoming ($I_h$)/reflected ($R_h$) solar radiations from the two Si-based pyranometers were compared to the hourly average of CNR1 measurements. The daily total incoming ($I_d$)/reflected ($R_d$) solar radiations for both pairs of Si-based pyranometers were calculated and their averages were compared with the total $I_d$/$R_h$ of CNR1. Albedo (α) for each pyranometer was calculated as the ratio of total $R_d$ and total $I_d$ radiation. The daily average α from the two pairs of Si-based pyranometers was compared to the α value from the CNR1 pyranometer. For performance comparisons, simple linear regression models were used, testing the hypotheses that the intercept of linear regression was not different from 0 and the slope not different from 1 (using the *linearHypothesis()* function of the R package "car" (Fox and Weisberg, 2011)). All analyses were conducted using the R statistical platform (The R Core Team, 2019). Graphs were created using the R-package 'ggplot2' (Wickham, 2016).

[Figure]

[Figure]

**Photo S1:** Silicon-based and CNR1 pyranometers measuring albedo at the Elora Research Station, Guelph, Ontario, Canada (Photo Credit: Shannon Brown, Postdoctoral Research Associate, School of Environmental Science, University of Guelph).

**S3. Results and Discussions**

Results from the simple linear regression of $I_h$ measured by CNR1 and Si-based pyranometers indicated a very close match between measurements ($R^2 = 0.985$, Residual Standard Error [RSE] = 28.05, $p < 0.01$) (Fig. S1a). The regression intercept (–1.36) was not significantly different from 0 ($p > 0.05$); as expected the slope (1.23) was significantly different from 1 ($p <$

Moved down [1]:

Moved (insertion) [1]

Deleted: First author after installing the pyranometers at the Elora Research Station, Guelph, Ontario, Canada (Photo Credit: Shannon Brown, Postdoctoral Research Associate, School of Environmental Science, University of Guelph).¶

[Figure]

**Figure S1.** Field comparisons of Si pyranometers (300–1100 nm) with thermopile pyranometers (305–2800 nm) under different sky and ground conditions over nine days. **a)** comparison of measured hourly irradiance ($I_h$). **b)** comparison measured hourly reflected radiation ($R_h$). **c)** comparison of measured daily total irradiance ($I_d$). **d)** comparison of measured daily total reflected radiance ($R_d$). **e)** comparison of measured daily albedo ($\alpha$). **f)** comparison of old vs. new Si-based pyranometers measurements of hourly irradiance and reflected radiation. RSE indicates Residual Standard Error.

0.01), reflecting additional measured energy flux at wavelengths >1100 nm. A similar strong linear relationship ($R^2 = 0.986$,

RSE = 20.05, p < 0.01) was also observed for $R_h$ measured by CNR1 and Si-based pyranometers (Fig. S1b). The regression intercept (2.64) was not significantly different from 0 (p > 0.05) and the slope (1.19) was significantly different from 1 (p < 0.01). The correspondence between pyranometers was higher for $R_h$ than it was for $I_h$ as indicated by higher $R^2$ and lower RSE. The correlation between measurements from the two types of pyranometers was even stronger when considered over a

80 24-hour period. For total $I_d$ the regression intercept was not significantly different from 0 (p > 0.05) and the slope (1.29) was not significantly different from 1 (p = 0.08) ($R^2 = 0.99$, RSE = 162.20, p < 0.01) (Fig. S1c). For total $R_d$ the regression intercept was also not significantly different from 0 (p > 0.05) and the slope (1.16) was not significantly different from 1 (p = 0.07) ($R^2 = 0.99$, RSE = 129.5, p < 0.01) (Fig. S1d).

85 Results from the simple linear regression for $I_h$ and $R_h$ measurements from the old and new Si-based pyranometers indicated exceptionally close correspondence ($R^2 = 0.99$, p < 0.01) (Fig. S2f). For $I_h$ the regression intercept (15.59) was significantly different from 0 (p = 0.03) and the slope (1.01) was not significantly different from 1 (p > 0.05). For $R_h$ the regression intercept (-5.92) was however not different from 0 (p > 0.05) and the slope (0.96) not different from 1 (p = 0.1).

90 Figure S1e indicates close agreement in daily albedo measurement between the CNR1 and Si-based ($R^2 = 0.93$, RSE = 0.04, p < 0.01). The regression intercept (0.016) of this relationship was not significantly different from 0 (p > 0.05) and the slope (0.98) was not significantly different from 1 (p > 0.05). The daily albedo difference between the CNR1 and Si-based pyranometers ranged from –0.0601 to 0.064, which was well within the previously reported acceptable (~5–7%) error range for class one pyranometers (Myers, 2010; Stroeve et al., 2005). Over the nine-day measurement period, the mean absolute dif-

95 ference in daily albedo was 0.037 (± 0.014), and the mean difference in average daily albedo was negligible (0.0028 ± 0.031). We did not find any detectable pattern in deviations between sensor types with increased/decreased cloud cover and ground snow cover. Since the difference in mean daily albedo values is negligible and the regression slope and intercept are not statistically different from 1 and 0, respectively, we conclude that albedo measurements of CNR1 and Si-based pyranometers used are closely comparable, and thus there is no need to perform any corrections on Si-based pyranometer meas-

100 urements.

**S4. References**

Dirmhirn, I.: On the use of silicon cells in meteorological radiation studies, Journal of Applied Meteorology, 7(4), 702–707, 1968.

105 Environment Canada: Historical climate data, [online] Available from: http://climate.weather.gc.ca/historical_data/search_historic_data_e.html (Accessed 4 March 2019), 2019.

Fox, J. and Weisberg, S.: An {R} Companion to Applied Regression, 2nd ed., Sage, Thousands Oaks, CA. [online] Available from: http://socserv.socsci.mcmaster.ca/jfox/Books/Companion, 2011.

110 François, C., Ottlé, C., Olioso, A., Prévot, L., Bruguier, N. and Ducros, Y.: Conversion of 400-1100 nm vegetation albedo measurements into total shortwave broadband albedo using a canopy radiative transfer model, Agronomie, 22(6), 611–618, doi:10.1051/agro:2002033, 2002.

115 Myers, D. R.: Comparison of direct normal irradiance derived from silicon and thermopile global hemispherical radiation detectors, edited by N. G. Dhere, J. H. Wohlgemuth, and K. Lynn, p. 77730G, San Diego, California., 2010.

Stroeve, J., Box, J. E., Gao, F., Liang, S., Nolin, A. and Schaaf, C.: Accuracy assessment of the MODIS 16-day albedo product for snow: comparisons with Greenland in situ measurements, Remote Sensing of Environment, 94(1), 46–60, doi:10.1016/j.rse.2004.09.001, 2005.

120

The R Core Team: R: A language and environment for statistical computing, R, R Foundation for Statistical Computing, Vienna, Austria., 2019.

125 Wickham, H.: ggplot2: Elegant Graphics for Data Analysis, R, Springer-Verlag, New York. [online] Available from: http://ggplot2.org, 2016.

130

135

140

145

150

155

**Supplementary Tables**

**Supplementary Table 1: Regression model coefficients and fit statistics for albedo as a function of stand attributes (without secondary data) in different seasons in the boreal forest**

| Season | Post-harvest stands | | | | Post-fire stands | | | |
|---|---|---|---|---|---|---|---|---|
| | *Parameter Estimates* | | *Model Fit* | | *Parameter Estimates* | | *Model Fit* | |
| | *Coefficient* | *Estimate* | *ΔAIC* | *Adj. R²* | *Coefficient* | *Estimate* | *ΔAIC* | *Adj. R²* |
| Winter | Intercept | **- 49.53** | **- 560.7** | **0.99** | Intercept | 0.056 | -18.2 | 0.84 |
| | SA | **11.99** | | | SA | - 0.111 | | |
| | PDBS | **0.926** | | | PDPS | -0.022 | | |
| | CH | **1.728** | | | SA:PDBS | 0.002 | | |
| | SA:CH | **- 0.659** | | | | | | |
| | SA:PDBS | **- 0.228** | | | | | | |
| Spring | Intercept | **- 7.195** | -495.4 | 0.99 | Intercept | **-1.747** | -18.8 | 0.92 |
| | SA | **1.298** | | | SA | 0.016 | | |
| | PDBS | **0.116** | | | PDBS | 0.002 | | |
| | CH | **- 1.264** | | | | | | |
| | SA:PDBS | **- 0.024** | | | | | | |
| Summer | Intercept | **-3.987** | -571.3 | 0.99 | Intercept | 6.591 | -289.8 | 0.97 |
| | SA | **0.176** | | | SA | -1.473 | | |
| | PDBS | **0.017** | | | PDBS | -0.142 | | |
| | GVC | **0.074** | | | CH | 2.379 | | |
| | SA:GVC | **-0.004** | | | SA:PDBS | 0.158 | | |
| | PDBS:GVC | **-0.001** | | | | | | |
| | SA:PDBS | **7.4e-05** | | | | | | |
| Fall | Intercept | **0.398** | -6.1 | 0.94 | $\frac{4.5}{6.87}\,e^{(-\frac{SA-13.2}{6.87}-e^{-\frac{SA-13.2}{6.87}})}$ | | -3.1 | 0.045[1] |
| | SA | 0.013 | | | | | | |
| | CH | **-0.182** | | | | | | |
| | GVC | **-0.007** | | | | | | |
| | SA:CH | **0.007** | | | $0.99\,e^{0.013\,PDBS}$ | | -25.4 | 0.008[1] |
| | CH:GVC | **0.005** | | | | | | |
| | SA:CH:GVC | **-0.0002** | | | | | | |
| | $\frac{28.86}{45.39}\,e^{(-\frac{PDBS-67.62}{45.39}-e^{-\frac{PDBS-67.62}{45.39}})}$ | | -0.9 | 0.049[1] | | | | |

**Notes:** SA, PDBS, CH, and GVC indicate stand age (year), proportion of deciduous broadleaf species (%), canopy height (m), and ground vegetation cover (%), respectively. Parameter estimates for GLMs in bold and regular fonts indicate statistical significance at 1% and 5% level, respectively. For fall nonlinear regression models, 28.86 and 45.39 coefficients of post-harvest stands were significant at 5% level and the rest is significant at 1% level. [1] indicates residual stand error of the nonlinear regression model. ΔAIC = AIC of the best-fit model – AIC of the corresponding null model. The goodness-of-fit of these models were compared against the corresponding null models (using deviance) and were found to be significantly better than the corresponding null models.

160

**Other Supplementary Figures**

[Figure]

**Supplementary Figure 2.** Relationships between stand age and stand structural properties in the winter season. Best-fit models were selected using an AIC-based algorithm from a set of candidate models. Estimated parameters of all models are significant at 5% level. ΔAIC = AIC of the best-fit model – AIC of the corresponding null model. RSE = Residual Standard Error of the best-fit model.

165

[Figure]

**Supplementary Figure 3.** Relationships between stand age and stand structural properties in the summer season. Best-fit models were chosen using an AIC-based algorithm from a set of candidate models. Estimated parameters of all models are significant at 5% level. ΔAIC = AIC of the best-fit model – AIC of the corresponding null model. RSE = Residual Standard Error of the best-fit model.

[Figure]

**Supplementary Figure 4.** Relationships between stand age and stand structural properties in the spring season. Best-fit models were chosen using an AIC-based algorithm from a set of candidate models. Estimated parameters of all models are significant at 5% level. ΔAIC = AIC of the best-fit model – AIC of the corresponding null model. RSE = Residual Standard Error of the best-fit model.

[Figure]

175 **Supplementary Figure 5.** Relationships between stand age and stand structural properties in the fall season. Best-fit models were chosen

using an AIC-based algorithm from a set of candidate models. Estimated parameters of all models are significant at 5% level. ΔAIC =

AIC of the best-fit model – AIC of the corresponding null model. RSE = Residual Standard Error of the best-fit model.